# FUSE: FK-Steered Multi-Modal Flow Matching for Efficient Simulation-Based Posterior Estimation

**Weichen Qin** [1]  **Yufan Xie** [1]  **Peihao Wang** [2]  **Chia-Jui Chou** [1]  **Minghui Du** [3 4]  **Peng Xu** [3 4]  **Ziren Luo** [3 4]  **Yi Yang** [1]
**Jingyi Yu** [1]  **Bo Liang** [3 4]  **Jiakai Zhang** [1 5]

## Abstract

Simulation-Based Inference (SBI) is critical for scientific discovery, with generative models offering a promising path toward efficient inference. However, existing methods struggle with effective multimodal modeling. They often rely on brute-force fusion strategies that ignore the structural disparities between parameters and observations, thus limiting estimation fidelity. In this work, we introduce FUSE (Feynman-Kac steered mUlti-modal flow matching for efficient Simulation-based posterior Estimation). Unlike prior work, FUSE employs a dual-track architecture that preserves the distinct features of multimodal inputs while facilitating dynamic interaction. Additionally, we propose an FK-steered sampling strategy that leverages intermediate observation likelihoods to guide the generative trajectories, effectively improving the sample quality during inference. Our approach outperforms state-of-the-art baselines on standard SBI benchmarks, producing posteriors that closely match reference MCMC posterior samples. Furthermore, in a real-world exoplanet orbital estimation task, FUSE successfully resolves complex parameter degeneracies that challenge existing methods, highlighting its potential to accelerate complex scientific discoveries in astrophysics and beyond.

[1]ShanghaiTech University, Shanghai, China [2]The University of Texas at Austin, Austin, TX, USA [3]Center for Gravitational Wave Experiment, National Microgravity Laboratory, Institute of Mechanics, Chinese Academy of Sciences, Beijing 100190, China [4]Taiji Laboratory for Gravitational Wave Universe (Beijing/Hangzhou), University of Chinese Academy of Sciences (UCAS), Beijing 100049, China [5]Cellverse, Co., Ltd. Correspondence to: Bo Liang <liangbo22@mails.ucas.ac.cn>, Jiakai Zhang <zhangjk@shanghaitech.edu.cn>.

*Proceedings of the $43^{rd}$ International Conference on Machine Learning*, Seoul, South Korea. PMLR 306, 2026. Copyright 2026 by the author(s).

## 1. Introduction

The rapid evolution of generative models (Kingma & Welling, 2014; Lipman et al., 2023; Ho et al., 2020) has recently unlocked transformative capabilities in multi-modal domains, enabling seamless translation between text, images, videos, and 3D assets (Zhang et al., 2024). Beyond their applications in entertainment, these powerful cross-modal transformation capabilities hold immense potential for scientific discovery, particularly in solving inverse problems. In such tasks, the objective is to infer the posterior distribution of latent parameters based on observed data challenge omnipresent in astrophysics, ranging from orbital parameter estimation (Liang et al., 2026b) to the characterization of gravitational wave sources from detectors either on the earth (Dax et al., 2021) or in space (Srinivasan et al., 2025; Du et al., 2024; Liang et al., 2024; 2026a; 2025a;b).

Despite their potential, estimating the posterior distribution in this context is non-trivial due to the intrinsic complexity of scientific data. First, a primary obstacle is the heterogeneous dimensionality between the input observations and the target parameters. Learning a robust mapping from high-dimensional sensor data to specific physical properties—often within a vast and complex search space, leading to a requirement for the customized design of model architectures. Second, the fidelity requirements are far more stringent than in general vision tasks; the model must accurately estimate the full posterior distribution, capturing multi-modal structural information and subtle correlations to avoid biased parameter estimation. Moreover, practical deployment demands high-throughput inference, requiring models that can generate accurate samples rapidly to keep pace with large-scale survey data or time-sensitive astronomical events.

Existing solutions face a fundamental trade-off between accuracy and efficiency. While traditional algorithms like Markov Chain Monte Carlo (MCMC) (Foreman-Mackey et al., 2013) provide asymptotically exact posteriors, they incur prohibitive computational costs, often requiring weeks for a single event. Neural approaches (Wildberger et al., 2023; Gonçalves et al., 2020) offer rapid inference but lack customized architectural designs, leading to suboptimal per-

formance. Specifically, FMPE (Wildberger et al., 2023) compresses observations into static embeddings and employs a brute-force fusion strategy. This approach ignores the structural disparity between time, parameters, and sensory data, significantly constraining effective multimodal generation. Consequently, these methods often fail to faithfully recover the complex posterior distribution in real-world applications.

To address these limitations, we propose a novel framework, FUSE, which adopts a **F**eynman-Kac steered m**U**ltimodal flow matching for efficient **S**imulation-based posterior **E**stimation. Inspired by the multi-modal diffusion transformer (MMDiT) (Esser et al., 2024), our model embeds parameters and observations in a dual-track framework to better preserve their distinct structural features, while enabling bidirectional interaction through a subsequent fusion module. This allows the generative flow to dynamically "revisit" the raw observation data at every integration step, ensuring maximal information extraction. Furthermore, to narrow the gap between fast amortized SBI and likelihood-based posterior correction, we introduce an FK-steered sampling strategy. Amortized generative models can place probability mass in physically implausible regions when the learned transport is imperfect, whereas asymptotically exact likelihood-based samplers are often too slow for time-sensitive scientific inference. FK-steering injects simulator likelihood information into the generative trajectory itself: by propagating multiple concurrent trajectories and resampling them at intermediate times, FUSE allocates more computation to high-density posterior regions. In the experiments below, this improves posterior fidelity on difficult inverse problems while adding only modest inference overhead relative to the amortized sampler.

In our experiments, we comprehensively validate our method on a widely used SBI benchmark (Lueckmann et al., 2021). Compared to state-of-the-art neural methods, our model produces posterior distributions that most closely match reference MCMC results across diverse scenarios, demonstrating superior approximation capabilities over MLP-based baselines. To evaluate the efficacy of our FK-steered sampling strategy, we compare performance with and without this component on the challenging SLCP task. Results show that FK-steered inference yields more compact and high-fidelity posteriors, provided by higher mean and peak likelihood scores. Subsequently, we apply our model to a challenging real-world scientific problem: exoplanet orbital parameter estimation. Leveraging our advanced architecture and sampling strategy, our method successfully resolves complex parameter degeneracies that existing SBI approaches may fail to produce a meaningful posterior distribution. We believe this work presents a scalable, high-precision framework for future scientific discovery.

## 2. Related Work

**Traditional posterior estimation.** Traditional likelihood-free inference approximates the Bayesian posterior without evaluating the likelihood by repeatedly simulating data and retaining (or weighting) parameters whose simulations match the observation under a prescribed discrepancy, typically defined on summary statistics with a tolerance threshold. Historically, this line of work progressed from *Rejection ABC*—a direct accept/reject procedure driven by discrepancy-based filtering (Tavaré et al., 1997)—to *ABC-MCMC*, which embeds the same acceptance criterion within a Metropolis–Hastings kernel to concentrate computation in higher-posterior regions (Marjoram et al., 2003). In parallel, *Synthetic Likelihood* methods introduced a parametric likelihood (often Gaussian) for summary statistics, enabling standard likelihood-based inference while retaining simulation as the primitive (Wood, 2010). Despite these advances, classical approaches often exhibit *poor mixing* in high-dimensional parameter spaces and are inherently *non-amortized*, requiring costly simulator calls for each new observation and each step of the chain. In contrast, our method leverages FK-steered flow matching to enable amortized inference.

**Neural posterior estimation.** Neural SBI has sought to modernize likelihood-free inference by pairing simulators with learned surrogates that improve practicality and scalability. One direction augments classical MCMC by amortizing likelihood(-ratio) information with neural estimators to drive Metropolis–Hastings transitions (Hermans et al., 2020) (and related direct ratio formulations that yield more informative transition signals) (Cobb et al., 2024). A second direction replaces chains altogether with amortized inference networks, using variational, adversarial, or score-/diffusion-based training to generate posterior samples directly (Glöckler et al., 2022; Ramesh et al., 2022; Geffner et al., 2023; Gloeckler et al., 2024). Despite progress, MCMC-coupled methods remain limited by mixing and per-step costs, while many MCMC-free approaches underexploit the multimodal structure of SBI when fusing parameters and observations, which can hinder MCMC-level fidelity and MAP-oriented estimation. Our method addresses both issues by adapting MMDiT-style joint-attention fusion to parameter–observation inference, using flow matching to learn an amortized multimodal transport, and leveraging inference-time scaling to more reliably sample high-quality parameter estimates.

**Multimodal generative models** Generative modeling has rapidly improved the quality-efficiency trade-off, with dominant paradigms evolving from latent-variable models (including multimodal VAEs) to score-based models and, more recently, flow-matching objectives that learn transport dy-

namics directly (Kingma & Welling, 2014; Suzuki et al., 2016; Wu & Goodman, 2018; Shi et al., 2019; Sutter et al., 2021; Ho et al., 2020; Song et al., 2021b; Lipman et al., 2023). Meanwhile, sampling has been steadily accelerated from ancestral DDPM to DDIM and modern fast samplers such as rectified flow and ODE/SDE solvers that deliver high-quality samples in far fewer steps (Ho et al., 2020; Song et al., 2021a; Liu, 2022; Lu et al., 2022; Karras et al., 2022; Salimans & Ho, 2022; Song et al., 2023). Architectures have also shifted from U-Nets to multimodal Diffusion transformer, which promote joint attention over modality tokens for stronger cross-modal interaction (Ronneberger et al., 2015; Peebles & Xie, 2023; Bao et al., 2023; Chen et al., 2024; Esser et al., 2024). On the conditioning side, guidance and control mechanisms have evolved from classifier(-free) guidance to explicit multimodal controls (e.g., edges/layouts/instructions) and personalization, supported by scalable tokenization and alignment (Dhariwal & Nichol, 2021; Ho & Salimans, 2022; Zhang et al., 2023; Li et al., 2023; Brooks et al., 2023; Ruiz et al., 2023; Oord et al., 2017; Esser et al., 2021; Radford et al., 2021). In contrast to text or images, parameter estimation involves more complex simulator-driven dependencies and multimodal structures. We handle these by adapting a dual-track fusion architecture and using inference-time scaling to better sample high-density regions.

# 3. Preliminaries

## 3.1. Problem Setting

We consider the inverse problem of estimating a set of parameters $\boldsymbol{\theta} \in \mathbb{R}^N$ given observed data $\mathbf{x} \in \mathbb{R}^M$. The goal is to compute the posterior distribution $p(\boldsymbol{\theta}|\mathbf{x})$, which, according to Bayes' theorem, is proportional to the product of the likelihood and the prior:

$$p(\boldsymbol{\theta}|\mathbf{x}) \propto p(\mathbf{x}|\boldsymbol{\theta})p(\boldsymbol{\theta}). \tag{1}$$

Standard approaches for estimating this posterior rely on Markov Chain Monte Carlo (MCMC) methods, such as the Metropolis-Hastings algorithm. These algorithms generate a sequence of samples that asymptotically converges to the target distribution. At each iteration, given the current state $\boldsymbol{\theta}$, a new candidate $\boldsymbol{\theta}'$ is drawn from a proposal distribution $q(\boldsymbol{\theta}'|\boldsymbol{\theta})$. The candidate is accepted with probability $\alpha(\boldsymbol{\theta}'|\boldsymbol{\theta})$, defined as:

$$\alpha(\boldsymbol{\theta}'|\boldsymbol{\theta}) = \min\left(1, \frac{p(\mathbf{x}|\boldsymbol{\theta}')p(\boldsymbol{\theta}')}{p(\mathbf{x}|\boldsymbol{\theta})p(\boldsymbol{\theta})} \frac{q(\boldsymbol{\theta}|\boldsymbol{\theta}')}{q(\boldsymbol{\theta}'|\boldsymbol{\theta})}\right). \tag{2}$$

While powerful, the efficiency of this process depends heavily on the computational cost of the likelihood function $p(\mathbf{x}|\boldsymbol{\theta})$, where $\mathbf{x} = \{\mathbf{x}_1, \ldots, \mathbf{x}_n\}$ is a dataset consisting of $n$ independent observations with $\mathbf{x}_i \in \mathbb{R}^{d_x}$. Assuming the measurement noise follows a multivariate Gaussian

distribution with covariance matrix $\boldsymbol{\Sigma}_i \in \mathbb{R}^{d_x \times d_x}$, the log-likelihood is explicitly given by the sum of squared Mahalanobis distances:

$$\log p(\mathbf{x}|\boldsymbol{\theta}) = -\frac{1}{2}\sum_{i=1}^{n}(\mathbf{x}_i - \mathbf{f}_i(\boldsymbol{\theta}))^\top \boldsymbol{\Sigma}_i^{-1}(\mathbf{x}_i - \mathbf{f}_i(\boldsymbol{\theta})) + C, \tag{3}$$

where $\mathbf{f}_i(\boldsymbol{\theta}) \in \mathbb{R}^{d_x}$ denotes the theoretical prediction generated by a forward model for the $i$-th observation. In high-dimensional parameter spaces, evaluating the forward model $\mathbf{f}(\boldsymbol{\theta})$ and computing these quadratic forms for every proposal often leads to significant computational bottlenecks.

## 3.2. Flow Matching

Flow matching (FM) is a generative modeling framework that transforms a simple prior distribution $p_1$ (e.g., $\mathcal{N}(0, I)$) into a complex data distribution $p_0$ by learning a deterministic transport map. This transformation is modeled as an ordinary differential equation (ODE) defined by a time-dependent vector field $v_t : \mathbb{R}^d \to \mathbb{R}^d$:

$$\frac{\mathrm{d}}{\mathrm{d}t}\boldsymbol{\theta}_t = v_t(\boldsymbol{\theta}_t), \quad \boldsymbol{\theta}_1 \sim p_1, \tag{4}$$

where $\boldsymbol{\theta}_t$ represents the trajectory of a sample at time $t \in [0, 1]$. The goal is to parameterize $v_t$ with a neural network $v_\phi(\boldsymbol{\theta}, t)$ such that the flow at $t = 0$ generates samples from the data distribution $p_0$.

To train $v_\phi$, we employ conditional flow matching (CFM), which regresses the vector field onto a target conditional path defined between a data sample $\boldsymbol{\theta}_0 \sim p_0$ and a noise sample $\boldsymbol{\theta}_1 \sim p_1$. We specifically use the linear interpolation path (optimal transport path), defined as $\boldsymbol{\theta}_t = (1-t)\boldsymbol{\theta}_0 + t\boldsymbol{\theta}_1$. This path implies a constant target velocity: $\dot{\boldsymbol{\theta}}_t = \boldsymbol{\theta}_1 - \boldsymbol{\theta}_0$. The model is trained to approximate this target velocity by minimizing the expected mean squared error:

$$\mathcal{L}_{\mathrm{FM}} = \mathbb{E}_{t,\boldsymbol{\theta}_0,\boldsymbol{\theta}_1}\left[\left\|v_\phi(\boldsymbol{\theta}_t, t) - (\boldsymbol{\theta}_1 - \boldsymbol{\theta}_0)\right\|^2\right], \tag{5}$$

where $t \sim \mathcal{U}(0, 1)$. During inference, we generate samples by drawing $\boldsymbol{\theta}_1 \sim p_1$ and numerically integrating Eq. (4) backwards from $t = 1$ to $t = 0$ using the learned field $v_\phi$.

# 4. Methodology

As illustrated in Figure 1, we propose a unified framework for efficient and precise parameter estimation, synergizing a multimodal architecture for deep context-parameter fusion with inference-time scaling for enhanced sampling. Specifically, we adopt a Multimodal Diffusion Transformer (MMDiT) to facilitate bidirectional information exchange between noisy parameter tokens and observational contexts (Section 4.1). We train the model using a conditional rectified flow objective to learn optimal transport trajectories

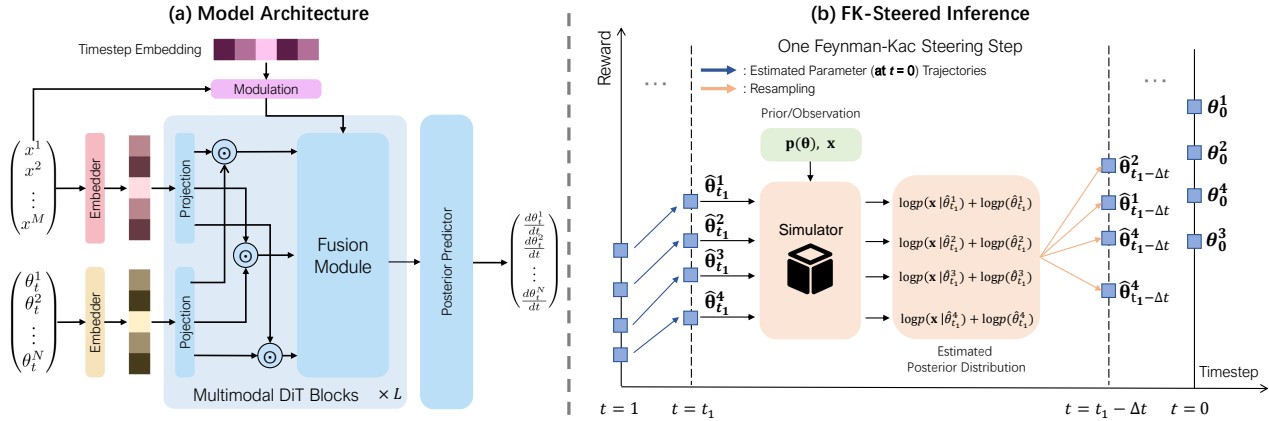

*Figure 1.* **Pipeline of FUSE. (a) FUSE's Architecture.** The architecture employs an independent embedding interface to map heterogeneous inputs into a shared space, followed by an MM-DiT-based fusion module for multimodal integration. Finally, the posterior predictor estimates the denoising velocity based on refined, parameter-wise token representations. **(b) FK-Steered Inference.** We steer the FUSE sampling process via a guidance mechanism that operates throughout the entire generative trajectory. By leveraging the *log-likelihood* from the simulator and the *prior density* of the parameters, the framework dynamically evaluates and rectifies the generative flow.

(Section 4.2). Finally, to address challenging scientific inverse problems, we introduce Feynman-Kac steered inference, which leverages test-time compute to dynamically navigate generative trajectories toward high-likelihood regions (Section 4.3). We kindly refer to Appendix B for more implementation details.

## 4.1. Model Architecture

**Input embedding.** To address the structural disparity between the physical parameters and sensory data, we design a specialized multi-modal embedding interface that projects these heterogeneous inputs into a shared latent space. Let $\boldsymbol{\theta} \in \mathbb{R}^N$ denote the unknown parameters and $\mathbf{x} \in \mathbb{R}^M$ the real observation. We represent both as high-dimensional embedding sequences with hidden width $D$. Specifically, we independently embed each scalar parameter $\theta_i$ into $K$ tokens via a simple MLP layer, yielding a sequence of parameter embeddings $\mathbf{f}_\theta^0 \in \mathbb{R}^{N_\theta \times D}$ where $N_\theta = N \times K$. Also, the observation is embedded into $N_x$ context tokens $\mathbf{f}_x^0 \in \mathbb{R}^{N_x \times D}$, where $N_x$ is a hyperparameter. To encode time information, we use a sinusoidal embedding $\mathbf{e}(t) \in \mathbb{R}^D$ together with a context projection to produce modulation parameters for adaptive normalization: a scaling vector $\boldsymbol{\gamma}(t, \mathbf{x}) \in \mathbb{R}^D$ and a shifting vector $\boldsymbol{\beta}(t, \mathbf{x}) \in \mathbb{R}^D$.

**Multimodal Diffusion Transformer.** To bridge the heterogeneous dimensionality of inputs, we adopt a dual-track architecture inspired by MM-DiT (Esser et al., 2024). At each layer $\ell$, the parameter tokens $\mathbf{f}_\theta^\ell$ and observation tokens $\mathbf{f}_x^\ell$ are first processed via modality-specific linear projections to preserve their distinct features. These projections are then concatenated within the joint self-attention mechanism:

$$[\mathbf{f}_\theta^{\ell+1}, \mathbf{f}_x^{\ell+1}] = \text{Attention}\left([\mathbf{Q}_\theta, \mathbf{Q}_x], [\mathbf{K}_\theta, \mathbf{K}_x], [\mathbf{V}_\theta, \mathbf{V}_x]\right). \tag{6}$$

where $\mathbf{Q}_m, \mathbf{K}_m, \mathbf{V}_m$ denote the projected queries, keys, and values for each modality $m \in \{\theta, x\}$. This mechanism enables parameter tokens to actively query relevant sensory features, capturing the cross-modal correlations necessary for high-fidelity posterior estimation. The resulting sequence is finally split back into $\mathbf{f}_\theta^{\ell+1}$ and $\mathbf{f}_x^{\ell+1}$ for subsequent layer-wise refinement. This iterative message passing refines the parameter representations layer-by-layer, effectively disentangling the multi-modal structures within the posterior distribution while preserving the specific token grouping size $K$.

**Posterior parameter predictor.** Upon obtaining the refined latent representations $\mathbf{f}_\theta^L$ from the final transformer layer, we decode the predicted flow velocity. We denote the feature sub-sequence for the $n$-th parameter as $\mathbf{f}_{\theta,n}^L \in \mathbb{R}^{K \times D}$. To map these high-dimensional features back to the scalar velocity domain, we employ a shared, lightweight linear readout head followed by a global aggregation mechanism:

$$v_{\phi,n}(\boldsymbol{\theta}_t, \mathbf{x}, t) = \frac{1}{K} \sum_{\mathbf{f} \in \mathbf{f}_{\theta,n}^L} \text{Linear}(\mathbf{f}). \tag{7}$$

Here, the linear layer projects each token independently, while the mean pooling synthesizes the distributed information into a robust scalar estimate. Finally, the full velocity vector is assembled as $\mathbf{v}_\phi = [v_{\phi,1}, \dots, v_{\phi,N}]^\top$. This decoupled readout strategy ensures that the complex joint dependencies between parameters are fully resolved within the deep transformer layers via self-attention. Thus, the final projection focuses solely on local feature extraction and dimensionality reduction.

## 4.2. Training Objective

We train a conditional rectified-flow velocity field $\mathbf{v}_\phi(\boldsymbol{\theta}_t, \mathbf{x}, t)$ with the standard RF regression loss. For each paired sample $(\mathbf{x}, \boldsymbol{\theta})$ with $\boldsymbol{\theta} \sim p_{\text{data}}(\boldsymbol{\theta} \mid \mathbf{x})$, draw $\boldsymbol{\epsilon} \sim \mathcal{N}(\mathbf{0}, \mathbf{I}_{d_\theta})$ and $t \sim \mathcal{U}(0, 1)$, and set $\boldsymbol{\theta}_t = t\,\boldsymbol{\epsilon} + (1-t)\,\boldsymbol{\theta}$. We minimize

$$\mathcal{L}(\phi) = \mathbb{E}\left[\left\|\mathbf{v}_\phi(\boldsymbol{\theta}_t, \mathbf{x}, t) - (\boldsymbol{\epsilon} - \boldsymbol{\theta})\right\|_2^2\right], \qquad (8)$$

where the expectation is over $(\mathbf{x}, \boldsymbol{\theta})$, $\boldsymbol{\epsilon}$, and $t$.

## 4.3. Feynman–Kac Steered Inference

**ODE sampler.** For efficient parameter inference, we solve the associated probability flow ordinary differential equation (PF-ODE) to transport samples from a simple base distribution to the posterior. We employ the first-order Euler method (Liu, 2022) to discretize the time horizon $t \in [1, 0]$ into $T$ uniform steps. Let $\{t_n\}_{n=0}^T$ denote the time sequence where $t_0 = 1$ and $t_T = 0$, with $\Delta t = -1/T$. The iterative update is

$$\boldsymbol{\theta}_{t_{n+1}} = \boldsymbol{\theta}_{t_n} + \mathbf{v}_\phi(\boldsymbol{\theta}_{t_n}, \mathbf{x}, t_n)\,\Delta t, \qquad (9)$$

where $\boldsymbol{\theta}_{t_0}$ is drawn from the base noise distribution and $\boldsymbol{\theta}_{t_T}$ is the generated posterior sample.

**FK steering.** In complex scientific inverse problems, approximation errors in the learned velocity field can leave amortized samples with excess mass in physically implausible regions. We therefore use Feynman–Kac (FK) steering (Del Moral, 2004; Chopin & Papaspiliopoulos, 2020; Singhal et al., 2025) as a likelihood-guided correction during inference. The role of this step is not to make a finite particle sampler equivalent to MCMC. Instead, it uses simulator likelihoods to bias the learned generative trajectory toward plausible posterior regions while preserving the speed of amortized flow matching. Concretely, we maintain $B$ parallel particles and resample them at selected intermediate times according to FK potentials.

To reduce particle depletion after resampling, we use a stochastic sampler (Liu et al., 2025). With $h = 1/T$ and $t_{n+1} = t_n - h$, the reverse-time update is written as

$$\boldsymbol{\theta}_{t-h} = \boldsymbol{\theta}_t - \left[\boldsymbol{v}_\phi(\boldsymbol{\theta}_t, \boldsymbol{x}, t) - \frac{\sigma_t^2}{2t}\left(\boldsymbol{\theta}_t + (1-t)\,\boldsymbol{v}_\phi(\boldsymbol{\theta}_t, \boldsymbol{x}, t)\right)\right] h \\ + \sigma_t \sqrt{h}\,\boldsymbol{\epsilon}, \qquad (10)$$

where $\boldsymbol{\epsilon} \sim \mathcal{N}(\mathbf{0}, \boldsymbol{I})$, $\sigma_t = \alpha\sqrt{t/(1-t)}$, and $\alpha$ controls the stochastic perturbation. In implementation, the update is evaluated away from the endpoints of $[0, 1]$, with the denominator $1 - t$ clipped by a small constant for numerical stability.

**Potentials from simulator-based scoring.** Given a simulator and a prior, we define the particle reward function $r_\phi(\boldsymbol{\theta}_t, t; \mathbf{x})$ as the unnormalized log-posterior of the clean parameter estimated from the current state. Specifically,

$$r_\phi(\boldsymbol{\theta}_t, t; \mathbf{x}) = \log p(\mathbf{x} \mid \hat{\boldsymbol{\theta}}_t) + \log p(\hat{\boldsymbol{\theta}}_t), \\ \text{with} \quad \hat{\boldsymbol{\theta}}_t = \boldsymbol{\theta}_t - t\,\mathbf{v}_\phi(\boldsymbol{\theta}_t, \mathbf{x}, t), \qquad (11)$$

where the sign follows the interpolation $\boldsymbol{\theta}_t = (1 - t)\theta_0 + t\epsilon$ and velocity target $\epsilon - \theta_0$. We evaluate the simulator at this denoised proxy rather than at the noisy intermediate state, since $\boldsymbol{\theta}_t$ need not satisfy the physical constraints of the simulator. The resulting potentials are best understood as a tractable likelihood-based tilt of the learned path measure; the finite-particle implementation is an approximate posterior-correction mechanism, not an asymptotically exact replacement for MCMC. A more detailed theoretical perspective is provided in Appendix I. We employ this reward to construct potentials $G_t$ that guide generation. Let $R = \{t_{r_1}, \ldots, t_{r_J}\}$ be the interval-resampling schedule; equivalently, non-resampling steps carry the neutral potential $G_t = 1$. At the $j$-th resampling time $t_{r_j}$, we use

$$G_{r_j}(\boldsymbol{\theta}_{t_{r_j}}) := \exp\left(\frac{\lambda}{J} r_\phi(\boldsymbol{\theta}_{t_{r_j}}, t_{r_j}; \mathbf{x})\right), \qquad (12)$$

where $\lambda$ is a global scaling factor and $1/J$ corresponds to a uniform annealing schedule. This follows the interval-resampling form of FK steering: particles are propagated by the learned sampler and are resampled using normalized potential scores. Since our proposal is the learned reverse transition itself, the transition ratio in the general FK importance weight reduces to one; an alternative proposal would require the corresponding $p_\theta/\tau$ correction.

**Resampling strategy.** To focus computation on high-likelihood regions without turning the sampler into a deterministic post-hoc selector, we resample particles at the predefined times $r_j$. The normalized weights are computed from the FK scores themselves:

$$w_{r_j}^k = \frac{G_{r_j}(\boldsymbol{\theta}_{t_{r_j}}^k)}{\sum_{\ell=1}^B G_{r_j}(\boldsymbol{\theta}_{t_{r_j}}^\ell)} = \text{softmax}_k\left(\frac{\lambda}{J} r_\phi(\boldsymbol{\theta}_{t_{r_j}}^k, t_{r_j}; \mathbf{x})\right). \qquad (13)$$

We then perform multinomial resampling *with replacement* to obtain the next particle population. High-score trajectories may be copied several times, while trajectories with negligible posterior reward are likely to be discarded. The stochastic propagation steps following resampling reintroduce local diversity, which empirically reduces the under-dispersion one would obtain from deterministic best-of-$N$ filtering.

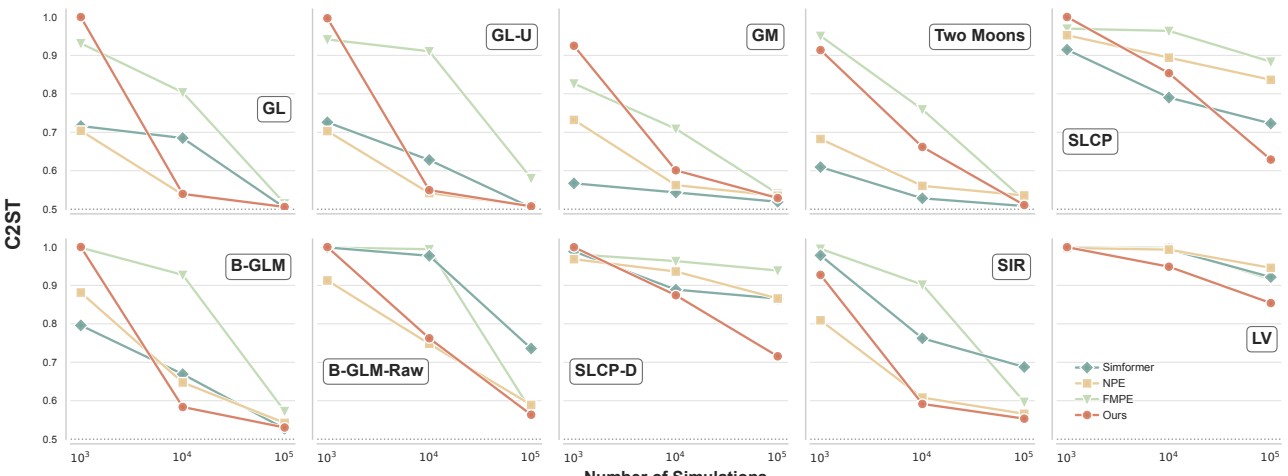

*Figure 2.* **Sample efficiency comparison on 10 SBIBM tasks.** We evaluate posterior fidelity using observation-wise $\ell$-C2ST across simulation budgets from $10^3$ to $10^5$, where 0.5 indicates an ideal match to the reference posterior. Curves report means over the official SBIBM observations, and shaded regions indicate the corresponding standard deviations. FUSE is evaluated without FK-steering in this benchmark to isolate the contribution of the MM-DiT architecture. While the high-capacity backbone is less competitive in the extremely low-budget regime on some tasks, it scales strongly with simulator coverage and reaches the best data-rich performance on the most challenging settings.

| METHOD | POSTERIOR FIDELITY | | STATISTICAL DISCREPANCY | | ESTIMATION ACCURACY | | |
|---|---|---|---|---|---|---|---|
| | $\ell$-C2ST ($\downarrow$) | MMD ($\downarrow$) | KL ($\downarrow$) | SINKHORN ($\downarrow$) | PME ($\downarrow$) | PVR ($\rightarrow 1$) | MEDDIST ($\downarrow$) |
| NPE | 0.64 | 0.046 | 0.89 | 0.228 | 0.19 | 9.14 | 1.60 |
| FMPE | 0.66 | 0.033 | 0.94 | 0.115 | 0.16 | 3.95 | 1.62 |
| SIMFORMER | 0.63 | 0.040 | 0.66 | 0.085 | 0.20 | 1.55 | 1.66 |
| FUSE | **0.59** | **0.016** | **0.28** | **0.077** | **0.06** | **1.27** | **1.55** |

*Table 1.* **Quantitative performance on the SBIBM benchmark.** All numbers are averaged over the 10 SBIBM tasks at the $10^5$ simulation budget. FUSE is evaluated without FK-steering to isolate the contribution of the MM-DiT architecture. We report observation-wise $\ell$-C2ST averaged over benchmark observations; close to 1 is better for PVR, lower is better for all other metrics. Best results are highlighted in **bold**.

## 5. Experiments

**Metrics.** We evaluate our method and baselines using seven standard metrics: local observation-wise Classifier Two-Sample Test ($\ell$-**C2ST**) (Linhart et al., 2023); Maximum Mean Discrepancy (**MMD**) (Gretton et al., 2012); and Posterior Mean Error (**PME**). We further assess the fidelity of posterior spreads using the Posterior Variance Ratio (**PVR**) and compute the Median Distance (**MEDDIST**) between predicted samples and the true parameters. To capture higher-order distributional discrepancies, we also report Kullback-Leibler (**KL**) divergence (Wang et al., 2009) and **Sinkhorn** distance (Cuturi, 2013). For $\ell$-C2ST, a value of 0.5 represents an optimal match for each fixed observation before averaging; for PVR, values closer to 1 indicate better variance calibration; lower values are better for the remaining discrepancy and error metrics. Further details on these metrics can be found in Appendix C.

## 5.1. Comparison on SBI benchmark

**Dataset.** We evaluate on the *Simulation-Based Inference Benchmark* (SBIBM), implemented in the open-source sbibm framework (Lueckmann et al., 2021). It provides a standardized suite of **10** simulation-based inference tasks spanning simple statistical toy models and more challenging scientific simulators. Across tasks, parameter dimensionalities range from $\dim(\theta) \in [2, 10]$ and observation dimensionalities from $\dim(x) \in [2, 100]$. For each task, sbibm ships 10 fixed observations and corresponding $10^4$ reference posterior samples for quantitative evaluation, enabling a direct evaluation of posterior samples through established statistical distances and calibration-focused metrics.

**Baselines.** We compare our approach against three state-of-the-art amortized inference baselines: (1) **NPE (NSF)** (Papamakarios & Murray, 2016; Durkan et al., 2019), a density-estimation standard utilizing Neural Spline Flows to model complex posteriors; (2) **FMPE** (Wildberger et al., 2023), a flow-matching baseline that typically employs

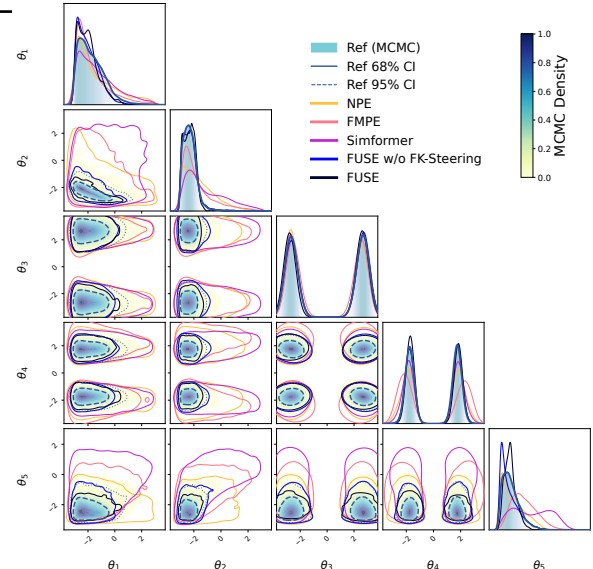

*Figure 3.* **FK-steering improves sample quality.** We compare our method against several baselines with the posterior samples and marginal distributions on the SLCP task. The addition of FK-steering further concentrates the samples into the high-density regions of the posterior.

ResNet-based vector field parameterization; and (3) **Simformer** (Gloeckler et al., 2024), a transformer-based framework leveraging self-attention to learn robust summary statistics. Since the official implementations of these baselines in the sbibm benchmark follow standardized hyperparameter protocols, we adopt their recommended settings to ensure a fair comparison. Note that we do not incorporate the FK-steering sampling strategy during evaluation by default to isolate the performance gains resulting from architectural differences alone.

**Evaluation protocol.** Unless otherwise stated, Table 1 and Figure 2 evaluate FUSE without FK-steering, so the comparison isolates the MM-DiT architecture from test-time likelihood guidance. Table 1 reports metrics at the $10^5$ simulation budget, averaged across the 10 SBIBM tasks and the official observations. The C2ST values are computed locally for each observation and then averaged, so we denote them as $\ell$-C2ST. Additional comparisons with SMC-ABC, NPSE, SNPE, SNLE, and task-wise budget analyses are provided in Appendix G.

**Quantitative Results.** We present a comprehensive quantitative analysis of the SBI benchmark in Table 1, where all reported metrics are averaged across the ten benchmark tasks. Our method consistently achieves strong performance across all seven evaluation metrics. In particular, it substantially improves distribution-level metrics such as MMD and KL divergence compared to existing baselines.

Overall, these results suggest that the proposed MM-DiT backbone provides a more effective mechanism for modeling complex parameter-observation relationships than standard MLP-based or existing transformer-based architectures. Additional benchmark evaluation details, task-wise statistics, simulation-budget analyses, and ablation studies are provided in Appendices D and G.

**Generalization capability.** Figure 2 shows the scalability and sample-efficiency behavior of our method. While all models improve as the simulation budget increases from $10^3$ to $10^5$, FUSE is not uniformly best in the extremely low-budget regime: the MM-DiT backbone has higher capacity and therefore requires sufficient simulator coverage to learn stable parameter-observation interactions. On low-difficulty tasks like Two Moons, GL, and GL-U, most methods eventually reach similar results near the ideal 0.5 threshold, suggesting that simple distributions can often be handled by lighter baselines given enough data. The advantage of FUSE becomes more pronounced on high-difficulty tasks such as LV, SLCP, and SLCP-D. In these challenging settings, which involve more complex posterior structures, existing baselines often plateau above the optimal $\ell$-C2ST score, whereas FUSE continues to improve with additional simulations. This supports the intended use case of FUSE: when offline simulator budgets are available, the high-capacity MM-DiT backbone can better exploit complex parameter-observation dependencies and deliver stronger data-rich posterior fidelity.

**FK-Steering improves sample quality.** To evaluate the effectiveness of the FK-Steering sampling strategy, we test our model on the challenging SLCP task. Further details on SLCP's score design can be found in Appendix A. As shown in Figure 3, even without the steering strategy, our model already provides a more accurate approximation of the reference posterior than the baselines. When we apply FK-Steering, the samples are further guided toward high-density regions of the posterior. This improves mode localization on SLCP, but it should not be read as a guarantee of tail coverage: a likelihood-tilted particle system can be penalized by C2ST if it concentrates on high-probability regions while covering less of the reference tails. We therefore interpret the FK result as evidence that trajectory-level likelihood guidance improves high-density posterior fidelity and point-estimation behavior in this setting. Additional quantitative and qualitative analyses of FK-steering are provided in Appendix H.

### 5.2. Case Study: Real-time Orbital Characterization of $\beta$ **Pictoris b**

We evaluate the performance of FUSE by applying it to a real-world orbital parameter estimation of $\beta$ **Pictoris b**, a benchmark system in exoplanet science (Lagrange et al., 2010). Located approximately 63 light-years away, this system hosts a young, massive gas giant orbiting within a circumstellar debris disk (Nielsen et al., 2020). Accurately

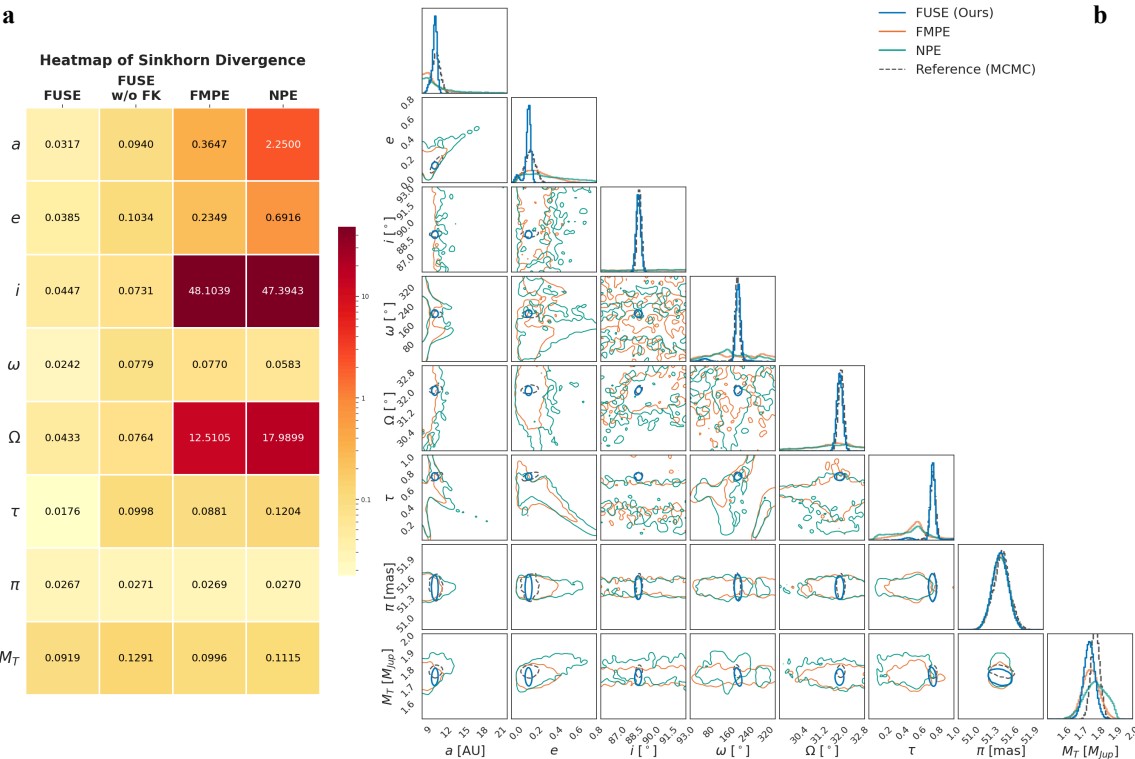

*Figure 4.* **Comprehensive evaluation of orbital parameter estimation for $\beta$ Pictoris b. (a) Quantitative comparison:** A heatmap of the Normalized Sinkhorn Divergence across all 8 orbital parameters. FUSE achieves the lowest error compared to baselines and the ablation version (w/o FK). Note the significant failure of NPE/FMPE in constraining orientation parameters $(i, \Omega)$, which our method resolves effectively. **(b) Posterior Reconstruction:** Marginalized posterior distributions for all parameters. The FUSE posteriors closely track the PTMCMC reference in high-probability regions, recovering sharp peaks and complex degeneracies that baselines miss.

constraining the orbit of $\beta$ Pictoris b is scientifically critical for understanding dynamic planet-disk interactions (Lacquement et al., 2025) and scheduling follow-up observations with instruments like JWST (Lightsey et al., 2012).

**Real-world astrometry and challenge.** The inference target is an 8-dimensional state space comprising the Keplerian elements and system properties, specifically total mass and parallax. This inverse problem is notoriously difficult because the available data often span only a small fraction of the total orbital period , leading to complex parameter covariances and highly non-Gaussian, multimodal posteriors (Blunt et al., 2020).

**Baselines.** We demonstrate the practical performance of FUSE by applying it to the orbital parameter estimation of $\beta$ Pictoris b. For this high-dimensional real-world task, we employ an expanded MM-DiT architecture with $D = 256$, $L = 12$, and $H = 8$. Detailed hyperparameter configurations, including the training schedule and specific FK-steering window settings, are provided in Appendix E. We benchmark FUSE against two state-of-the-art simulation-based inference methods: **NPE** (Tejero-Cantero et al., 2020; Ruth, 2024) and **FMPE** (Wildberger et al., 2023), as aforementioned. To ensure a fair comparison, all estimators are

trained on an identical budget of simulated orbits. We then evaluate their ability to generalize from the simulation before the real observational likelihood. In this real-world case, FUSE uses FK-steering sampling strategy to enhance result quality. As a reference, we employ a long-run Parallel Tempered MCMC (PTMCMC) chain (Vousden et al., 2016), which is designed to asymptotically sample the target posterior under standard convergence assumptions but requires substantial computational resources.

**Results.** We first quantitatively assess the estimation quality using the Normalized Sinkhorn Divergence between the predicted marginals and the PTMCMC reference. As shown in the heatmap (Figure 4a), FUSE consistently yields the lowest divergence scores (indicated by the lightest colors) across all physical parameters. In contrast, baselines suffer from catastrophic errors in orientation parameters (e.g., $i, \Omega$), marked by high divergence values (red blocks), whereas FUSE maintains high fidelity across the entire parameter space. Qualitatively, the corner plot (Figure 4b) visually corroborates these metrics. The FUSE contours closely align with the high-probability modes of the PTMCMC chain. Notably, our method successfully resolves the sharp peaks of the inclination $(i)$ and the longitude of the ascending node $(\Omega)$, whereas baseline methods exhibit sig-

nificant dispersion. The FUSE posteriors provide sharper constraints than the baselines, reflecting the integrated FK module's ability to filter out low-probability sampling noise and focus the estimate on the high-likelihood manifold. An ablation against FUSE without FK-steering confirms that this improvement is not merely due to the amortized backbone. The unsteered model remains more mass-covering and less accurate around sharp degeneracies, whereas FK-steering improves both 95% credible-region overlap and mode accuracy: FUSE improves the posterior overlap from 0.52 to 0.62 and reduces the Mode L2 Distance from 6.60 to 3.85, while FMPE and NPE obtain substantially worse Mode L2 distances of 132.41 and 75.74, respectively. Full quantitative details are given in Appendix H.1. We further compare FUSE with a Naive Best-of-N selection strategy to show that applying likelihood and prior scores along the trajectory gives better high-density posterior fidelity on complex degeneracies than selecting only at the final step (see Appendix F for details).

**Efficiency: Towards Real-time Astronomy.** In terms of computational efficiency, while the reference PTMCMC chain typically requires hours (i.e., computation took $\approx 8.5$ hours using a 128-core CPU) to converge due to millions of sequential likelihood calculations, FUSE is capable of completing high-precision orbital inference within three minutes. The FK correction uses a small particle ensemble and lightweight simulator-based scoring, so its overhead remains modest relative to the cost of retraining or running a long observation-specific MCMC chain. A scaling breakdown over particle count and scoring frequency is reported in Appendix E.1. This dramatic acceleration enables "on-the-fly" orbital updating, allowing astronomers to refine ephemerides instantaneously as new astrometric data points are received.

## 6. Discussion

**Limitations.** Despite its high fidelity, FUSE, as an amortized inference method, theoretically lacks the asymptotic exactness guarantees of MCMC, and the implemented finite-particle FK-steering sampler should be interpreted as a tractable likelihood-guided correction rather than an exact replacement for MCMC. Second, the MM-DiT backbone is data-hungry: when simulator calls are extremely scarce, lighter amortized models or sequential observation-specific methods may remain preferable. Third, while FK steering mitigates approximation errors and improves high-density fidelity, accurately capturing extremely low-probability tails remains non-trivial. Finally, the scalability of FUSE from single-planet systems (e.g., $\beta$ Pictoris b) to higher-dimensional problems characterized by complex interactions—such as multi-planet systems or ground-based gravitational wave detection (e.g., LIGO, Virgo, KAGRA)—

remains to be empirically validated. Because the MM-DiT backbone and FK-steering module are modular, a promising future direction is to adapt FUSE to sequential or observation-specific SBI, where the learned multimodal transport initializes posterior proposals and FK likelihood steering prunes implausible regions during iterative refinement.

**Conclusion.** We have introduced FUSE, a framework integrating multi-modal flow matching with Feynman-Kac steering. By combining a dual-track architecture with likelihood-guided particle steering, FUSE improves the fidelity-efficiency trade-off of amortized SBI. In the $\beta$ Pictoris b experiment, it recovers the dominant PTMCMC posterior structure at substantially lower inference cost under the reported metrics. Improving robustness to low-SNR observations and non-stationary artifacts remains an important step toward broader astrophysical applications, including gravitational wave astronomy.

## Acknowledgements

This work was supported in part by the National Natural Science Foundation of China under Grant W2431046, the National Key R&D Program of China under Grant 2025YFA1309603, the Central Guided Local Science and Technology Foundation of China under Grant YDZX20253100001001, the Program for Grand Challenges in Basic Research under Grant No. 025GJHZ2025073GC, the MoE Key Lab of Intelligent Perception and Human-Machine Collaboration (ShanghaiTech University), and the Shanghai Frontiers Science Center of Human-centered Artificial Intelligence.

## Impact Statement

This paper introduces a Flow Matching framework designed to solve complex inverse problems, with a specific focus on applications in astrophysics. By enabling rapid and accurate posterior estimation from high-dimensional observational data, our work has the potential to significantly accelerate scientific discovery and maximize the utility of large-scale astronomical surveys.

However, applying generative models to scientific inference carries inherent risks. If the model is not properly calibrated or if the simulation-to-real gap is significant, there is a risk of generating biased posteriors or "hallucinated" physical parameters, which could lead to incorrect scientific conclusions. Therefore, practitioners must rigorously validate these models against domain-specific baselines before deployment.

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

# A. Exact log-posterior scores for SLCP

This appendix specifies the *score functions* used to evaluate candidate parameters via an unnormalized log-posterior. We consider the SBIBM task SLCP. The score is designed to be an *exact* evaluation of

$$\log p(\theta \mid x_{\text{obs}}) = \log p(\theta) + \log p(x_{\text{obs}} \mid \theta) - \log p(x_{\text{obs}}), \tag{14}$$

and we drop the constant $\log p(x_{\text{obs}})$ w.r.t. $\theta$.

## A.1. Generic posterior objective and the score used in sampling

Given an observation $x_{\text{obs}}$, Bayesian inference targets

$$p(\theta \mid x_{\text{obs}}) \propto p(\theta)\, p(x_{\text{obs}} \mid \theta). \tag{15}$$

A common objective for posterior maximization (MAP) is

$$\theta_{\text{MAP}} \in \arg\max_{\theta}\ \log p(\theta) + \log p(x_{\text{obs}} \mid \theta). \tag{16}$$

Likewise, MCMC methods (e.g., Metropolis–Hastings) only require an *unnormalized* log target:

$$\mathcal{S}(\theta; x_{\text{obs}}) := \log p(\theta) + \log p(x_{\text{obs}} \mid \theta), \tag{17}$$

since $\log p(x_{\text{obs}})$ cancels in acceptance ratios:

$$\log \alpha(\theta \to \theta') = \mathcal{S}(\theta'; x_{\text{obs}}) - \mathcal{S}(\theta; x_{\text{obs}}) + \log q(\theta \mid \theta') - \log q(\theta' \mid \theta). \tag{18}$$

We therefore implement task-specific $\log p(x_{\text{obs}} \mid \theta)$ and combine it with the task prior $\log p(\theta)$ to form $\mathcal{S}$.

## A.2. SLCP score

**Forward simulator.** SLCP uses a 5D parameter $\theta = (\theta_1, \theta_2, \theta_3, \theta_4, \theta_5) \in \mathbb{R}^5$ and produces $N = 4$ i.i.d. 2D observations. Define the transformed quantities

$$\mu(\theta) = (\theta_1, \theta_2)^\top \in \mathbb{R}^2, \tag{19}$$

$$s_1(\theta) = \theta_3^2, \qquad s_2(\theta) = \theta_4^2, \tag{20}$$

$$\rho(\theta) = \tanh(\theta_5) \in (-1, 1), \tag{21}$$

and the covariance matrix

$$\Sigma(\theta) = \begin{pmatrix} s_1(\theta)^2 & \rho(\theta)\, s_1(\theta)\, s_2(\theta) \\ \rho(\theta)\, s_1(\theta)\, s_2(\theta) & s_2(\theta)^2 \end{pmatrix} + \varepsilon I_2, \qquad \varepsilon > 0. \tag{22}$$

The simulator draws

$$x_n \mid \theta \overset{\text{i.i.d.}}{\sim} \mathcal{N}\big(\mu(\theta), \Sigma(\theta)\big), \qquad n = 1, \ldots, N, \tag{23}$$

and returns $x = \{x_n\}_{n=1}^N$, optionally flattened.

**Exact log-likelihood.** Given $x_{\text{obs}} = \{x_{\text{obs},n}\}_{n=1}^N$, the SLCP likelihood factorizes:

$$\log p(x_{\text{obs}} \mid \theta) = \sum_{n=1}^N \log \mathcal{N}\big(x_{\text{obs},n}; \mu(\theta), \Sigma(\theta)\big). \tag{24}$$

This is evaluated analytically via a multivariate normal density.

**Prior and score.** SLCP uses an independent uniform prior (as implemented by the benchmark):

$$p(\theta) = \prod_{d=1}^5 \text{Unif}(\theta_d; -3, 3), \quad \Rightarrow \quad \log p(\theta) = \sum_{d=1}^5 \log \text{Unif}(\theta_d; -3, 3). \tag{25}$$

The score is then

$$\mathcal{S}_{\text{SLCP}}(\theta; x_{\text{obs}}) = \log p(\theta) + \log p(x_{\text{obs}} \mid \theta), \tag{26}$$

with $\mathcal{S}_{\text{SLCP}}(\theta; x_{\text{obs}}) = -\infty$ if $\theta$ is outside the prior support or if numerical checks fail.

**Implementation notes.** We (i) reshape flattened observations to $[B, N, 2]$ for a batch of $B$ candidates, (ii) build $\Sigma(\theta)$ as in (22) with a small diagonal jitter $\varepsilon$ to guarantee positive semi definiteness, and (iii) sum per-point log densities as in (24). The score is computed in the *physical* parameter space; if candidates are produced in a standardized space, we first invert the standardization map before evaluating $\mathcal{S}_{\mathrm{SLCP}}$.

## B. Implementation details

We evaluate on the SBIBM benchmark using a unified lightweight Multi-Modal Diffusion Transformer (**MM-DiT**) with $D = 128, L = 6, H = 4$ by default. Motivated by their distinct physical semantics, our architecture treats parameters $\theta$, context $x$, and time $t$ as separate modalities rather than a concatenated vector. Context $x$ is projected to 4 tokens. For $\theta$, we employ *individual parameter tokenization*, mapping each scalar dimension to 2 tokens (reduced to 1 for the simpler *Two Moons* and *Gaussian* tasks). Training utilizes $100,000$ simulations via Adam (batch 512, cosine annealing schedule $2 \times 10^{-4} \rightarrow 1 \times 10^{-6}$) for 100 epochs. Basic Inference uses an Euler ODE solver with $N = 200$ steps. For FK-steered inference on SLCP task, we maintain a beam width $B$ of 8 particles. Resampling is performed every 5 steps within the window of step 20 to 200, with a noise scale $\alpha = 0.3$.

## C. Detailed Definitions of Metrics

In this section, we provide the formal mathematical definitions for the seven metrics used to evaluate the performance of our method and the baselines. In the following definitions, we assume a set of $N$ samples generated by the model, $\{\hat{\theta}_i\}_{i=1}^{N}$, and a set of $N$ reference posterior samples, $\{\theta_i\}_{i=1}^{N}$.

**Classifier Two-Sample Test (C2ST).** C2ST measures the similarity between two distributions by training a binary classifier to distinguish between generated and reference posterior samples. The dataset is divided into training and testing sets with equal representation from both distributions. The metric is defined as the classification accuracy on the test set:

$$\text{C2ST} = \frac{1}{N_{\text{test}}} \sum_{j=1}^{N_{\text{test}}} \mathbb{I}(\hat{y}_j = y_j), \tag{27}$$

where $\mathbb{I}$ is the indicator function, $y_j$ is the true label, and $\hat{y}_j$ is the predicted label. An ideal match results in an accuracy of 0.5, indicating that the classifier cannot distinguish between the two sets.

**Maximum Mean Discrepancy (MMD).** MMD quantifies the distance between two distributions $P$ and $Q$ by comparing their embeddings in a Reproducing Kernel Hilbert Space (RKHS). Given a kernel function $k(\cdot, \cdot)$, the squared MMD is defined as:

$$\text{MMD}^2(P, Q) = \mathbb{E}_{\theta, \theta' \sim P}[k(\theta, \theta')] + \mathbb{E}_{\hat{\theta}, \hat{\theta}' \sim Q}[k(\hat{\theta}, \hat{\theta}')] - 2\mathbb{E}_{\theta \sim P, \hat{\theta} \sim Q}[k(\theta, \hat{\theta})]. \tag{28}$$

In our implementation, we use a multi-scale Gaussian kernel following the standard protocol in the `sbibm` benchmark.

**Posterior Mean Error (PME).** PME evaluates the accuracy of the posterior mean by calculating the $L_2$ distance between the average of the predicted samples and the average of the reference posterior samples:

$$\text{PME} = \|\mathbb{E}[\hat{\theta}] - \mathbb{E}[\theta]\|_2. \tag{29}$$

This metric reflects the bias of the model in identifying the center of the posterior distribution.

**Posterior Variance Ratio (PVR).** PVR assesses how well the model captures the spread or uncertainty of the posterior. It is defined as the ratio of the predicted variance to the reference posterior variance, averaged over all parameter dimensions $d$:

$$\text{PVR} = \frac{1}{D} \sum_{d=1}^{D} \frac{\text{Var}(\hat{\theta}_d)}{\text{Var}(\theta_d)}. \tag{30}$$

A PVR value close to 1 indicates that the predicted distribution has a similar scale to the reference posterior.

**Median Distance (MEDDIST).** MEDDIST provides a robust measure of the distance between the generated samples and the true parameters $\theta^*$ used to generate the observation:

$$\text{MEDDIST} = \text{median}(\|\hat{\theta}_i - \theta^*\|_2). \tag{31}$$

Here, $\theta^*$ denotes the simulator ground-truth parameter associated with the observation when it is available. This metric captures the typical error of the predicted samples in the parameter space.

**Kullback-Leibler (KL) Divergence.** KL divergence measures the information discrepancy between the true distribution $P$ and the predicted distribution $Q$:

$$D_{KL}(P\|Q) = \int p(\theta) \log \frac{p(\theta)}{q(\theta)} d\theta. \tag{32}$$

Since the density $q(\theta)$ is typically not available in a closed form for generative models, we use a $k$-nearest neighbor estimator to calculate this value directly from the samples.

**Sinkhorn Distance.** Sinkhorn distance provides a geometric measure of the discrepancy between the predicted and reference distributions by approximating the optimal transport cost. We first compute the cost matrix $C$ using the squared Euclidean distance between the predicted samples $\hat{\theta}$ and the reference samples $\theta$, such that $C_{ij} = \|\hat{\theta}_i - \theta_j\|_2^2$. To ensure numerical stability across different tasks, we normalize $C$ by the maximum squared distance found within the reference samples. The distance is then obtained by solving an entropy-regularized optimal transport problem:

$$S_\varepsilon = \sum_{i,j} P_{ij} C_{ij}, \tag{33}$$

where $P$ is the optimal coupling matrix and $\varepsilon$ is the regularization parameter. In our implementation, we solve for $P$ using Sinkhorn iterations in the log-domain, which prevents numerical underflow and ensures more stable convergence during the optimization process.

## D. Ablation Study

We conduct a systematic ablation study on the SLCP task to evaluate the impact of different architectural choices. The SLCP task is chosen for this analysis because its complex posterior structure effectively highlights the performance differences between configurations. All experiments are performed on a single H20 GPU using a fixed budget of $10^5$ simulations and trained for 100 epochs.

| $D$ | Individual | Tokens/$\theta$ | Time | C2ST | KL |
|-----|-----------|-----------------|------|------|-----|
| 128 | False | 1 | 9.37 | 0.6734 | 0.3711 |
| 128 | True | 1 | 9.28 | 0.6458 | 0.3547 |
| 128 | True | 4 | 20.32 | 0.6581 | 0.4336 |
| 64 | True | 2 | **6.28** | 0.6553 | 0.3765 |
| 256 | True | 2 | 32.49 | 0.7583 | 0.8721 |
| 128 | True | 2 | 12.48 | **0.6336** | **0.2820** |

*Table 2.* Ablation study on the SLCP task. We analyze the influence of the hidden dimension ($D$), the individual embedding strategy (Individual), and the number of tokens per parameter (Tokens/$\theta$). We report the average inference time for $10^4$ samples(Time), C2ST, and KL divergence. The optimal configuration in the final row achieves the best balance between inference efficiency and posterior quality. Best results are highlighted in bold.

As shown in Table 2, we first compare embedding strategies. Mapping each parameter component individually (Individual=True) consistently yields better C2ST and KL scores than shared embeddings. For token allocation (Tokens/$\theta$), using 2 tokens per parameter provides the best efficiency; increasing this to 4 tokens significantly slows down inference without improving accuracy. Regarding the hidden dimension ($D$), the 128-dimensional setting proves optimal. While $D = 64$ is faster, it lacks the capacity for accurate estimation, and $D = 256$ leads to overfitting and higher computational costs. Based on these findings, we select $D = 128$, individual embeddings, and 2 tokens per parameter as our default configuration.

## E. Implementation details of Real-time Orbital Characterization of $\beta$ Pictoris b

We evaluate on Real-time Orbital Characterization of $\beta$ Pictoris b using a unified large Multi-Modal Diffusion Transformer with $Hidden dimensions = 256, Layers = 12, Number heads = 8$ by default. Motivated by their distinct physical semantics, our architecture treats parameters $\theta$, context $x$, and time $t$ as separate modalities rather than a concatenated vector. Context $x$ is projected to 12 tokens. For $\theta$, we employ *individual parameter tokenization*, mapping each scalar dimension to 1 tokens. Training utilizes $800,000$ simulations via Adam (batch 4096, cosine annealing schedule $5 \times 10^{-4} \to 1 \times 10^{-6}$) for 200 epochs. Basic inference uses an Euler ODE solver with $N = 200$ steps. For FK-steered inference, we maintain a beam width $B$ of 8 particles. Resampling is performed every 5 steps within the window of step 20 to 200, with a noise scale $\alpha = 0.3$.

### E.1. Inference-Time Compute and FK Overhead

The orbital-characterization experiment uses a compact FK configuration: $B = 8$ particles, resampling every 5 steps in the window from step 20 to step 200, and noise scale $\alpha = 0.3$. This adds simulator-based scoring during inference, but the scoring calls are lightweight compared with a long observation-specific PTMCMC chain. In our reported setting, PTMCMC requires approximately 8.5 hours on a 128-core CPU, whereas FUSE completes high-precision inference within three minutes under our setup.

| Particles | FK-scoring steps | Time |
|:---:|:---:|:---:|
| 8 | 17 | 152.77s |
| 8 | 81 | 172.02s |
| 4 | 17 | 71.11s |
| 16 | 17 | 278.20s |

*Table 3.* Inference-time overhead of FK-steering on the orbital-characterization experiment. Increasing the particle count has the dominant effect on runtime, while increasing the number of FK-scoring steps from 17 to 81 introduces a smaller additional cost in this setup.

These measurements indicate that, for this simulator and implementation, FK-steering scales primarily with the number of particles, while the simulator-based scoring frequency contributes a smaller overhead. This makes the chosen FK configuration practical for the reported real-time orbital-characterization setting.

## F. Comparison with Naive Best-of-N Selection

To further demonstrate the efficacy of our FK steered sampling, we compare FUSE against a "Naive Best-of-N" baseline. In this baseline, $N$ independent samples are generated using the standard flow-matching decoder, and only the sample with the highest likelihood (or lowest discrepancy) according to the simulator is selected at the final step. It is important to note that such a post-generation selection strategy is generic and could be readily applied to existing methods like NPE and FMPE.

As illustrated in Figure 5, while the Naive Best-of-N approach can identify high-probability candidates, it often fails to capture the intricate topological structures and sharp degeneracies of the reference posterior. In contrast, FUSE uses FK-steering to apply likelihood guidance at multiple intermediate timesteps. This trajectory-level correction concentrates computation on high-density regions more effectively than a single final selection. The results indicate that dynamic steering is more closely aligned with the MCMC reference in the reported metrics, particularly for complex parameter dependencies where static selection remains suboptimal.

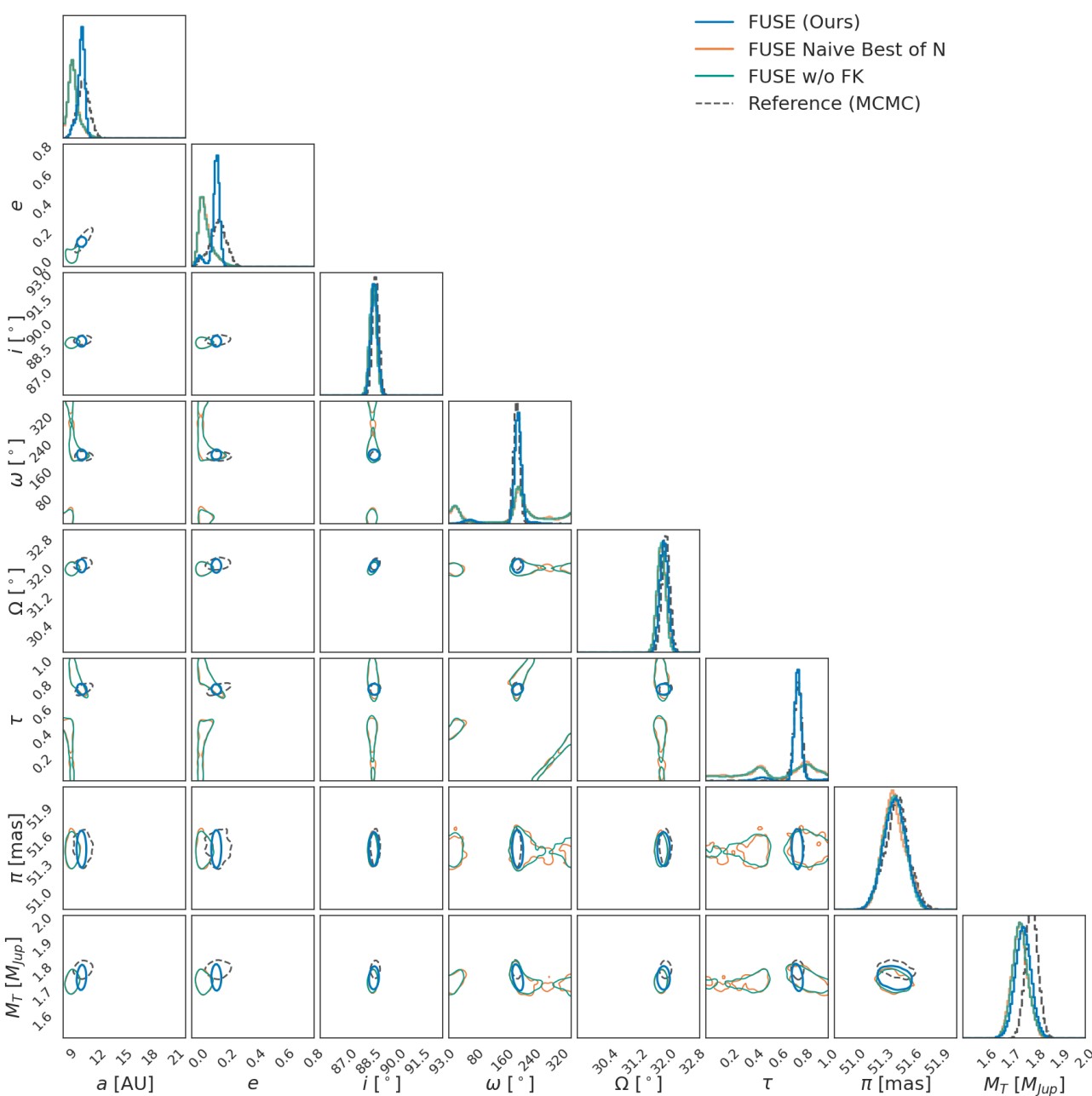

*Figure 5.* **Posterior distribution comparison between FUSE and Naive Best-of-N selection.** The corner plot displays the marginalized 1D and 2D posteriors for the $\beta$ Pictoris b orbital parameters. The Naive Best-of-N strategy (orange) performs selection only at the final generation step and does not recover several sharp degeneracies and correlations present in the reference. In contrast, **FUSE** (blue) uses sequential FK-steering to guide trajectories during generation, yielding a posterior that more closely follows the **MCMC reference** (black dashed/grey) in high-probability regions.

# G. Additional Benchmark Evaluation Details

## G.1. Evaluation Protocol

We provide additional implementation and evaluation details for the SBIBM benchmark experiments. All experiments follow the official SBIBM evaluation protocol introduced by Lueckmann et al. (2021).

For Table 1 in the main paper, all reported metrics are computed at the $10^5$ simulation budget and averaged across the 10

official observations provided by SBIBM for each task. The reported standard deviations correspond to the variability across these official observations.

For the SBIBM benchmark experiments, we disable FK-steering and evaluate FUSE purely as an amortized flow-matching posterior estimator. This setting isolates the effect of the proposed MM-DiT architecture from the additional inference-time guidance introduced by FK-steering.

In addition, the originally reported C2ST metric was computed independently for each observation and averaged afterward, which corresponds to the local classifier two-sample test metric ($\ell$-C2ST). To avoid ambiguity, we explicitly denote this metric as $\ell$-C2ST in the revised manuscript.

### G.2. Stability Across Official Observations

Table 4 reports task-wise $\ell$-C2ST results at the $10^5$ simulation budget. Results are reported as mean $\pm$ standard deviation across the 10 official SBIBM observations. We select SIR, SLCP, and LV as representative tasks covering different posterior complexities and dynamical behaviors.

| Method | SIR | SLCP | LV |
|---|---|---|---|
| Simformer | $0.6878 \pm 0.0533$ | $0.7229 \pm 0.0488$ | $0.9040 \pm 0.0493$ |
| FMPE | $0.5971 \pm 0.0429$ | $0.8836 \pm 0.0443$ | $0.9162 \pm 0.0495$ |
| NPE | $0.5661 \pm 0.0322$ | $0.8364 \pm 0.0430$ | $0.9456 \pm 0.0449$ |
| SMC-ABC | $0.6244 \pm 0.0322$ | $0.9589 \pm 0.0222$ | $0.9950 \pm 0.0034$ |
| NPSE | $0.5653 \pm 0.0204$ | $0.6949 \pm 0.0980$ | $0.8718 \pm 0.0699$ |
| SNPE | $\mathbf{0.5506 \pm 0.0238}$ | $0.6656 \pm 0.0608$ | $0.9281 \pm 0.0796$ |
| SNLE | $0.5847 \pm 0.0331$ | $\mathbf{0.5781 \pm 0.0293}$ | $\mathbf{0.6951 \pm 0.1025}$ |
| FUSE w/o FK | $0.5534 \pm 0.0122$ | $0.6290 \pm 0.0468$ | $0.8540 \pm 0.0888$ |

*Table 4.* Task-wise $\ell$-C2ST results at the $10^5$ simulation budget. Results are reported as mean $\pm$ standard deviation across the 10 official SBIBM observations. Lower values indicate better posterior fidelity.

### G.3. Performance Across Simulation Budgets

We further analyze $\ell$-C2ST on the challenging SLCP task under different simulation budgets. FUSE exhibits reduced performance in the extremely low-data regime, which is consistent with the larger model capacity of the MM-DiT architecture. However, its performance improves steadily as the simulation budget increases, leading to competitive posterior fidelity at larger scales.

| Method | $10^3$ | $10^4$ | $10^5$ |
|---|---|---|---|
| Simformer | $\mathbf{0.9147 \pm 0.0281}$ | $0.7902 \pm 0.0549$ | $0.7229 \pm 0.0488$ |
| FMPE | $0.9695 \pm 0.0148$ | $0.9636 \pm 0.0199$ | $0.8836 \pm 0.0443$ |
| NPE | $0.9524 \pm 0.0203$ | $0.8940 \pm 0.0420$ | $0.8364 \pm 0.0430$ |
| SMC-ABC | $0.9771 \pm 0.0097$ | $0.9624 \pm 0.0163$ | $0.9589 \pm 0.0222$ |
| NPSE | $0.9695 \pm 0.0157$ | $0.8255 \pm 0.0689$ | $0.6949 \pm 0.0980$ |
| SNPE | $0.9652 \pm 0.0194$ | $0.8445 \pm 0.0451$ | $0.6656 \pm 0.0608$ |
| SNLE | $0.9206 \pm 0.0251$ | $\mathbf{0.7131 \pm 0.0524}$ | $\mathbf{0.5781 \pm 0.0293}$ |
| FUSE w/o FK | $0.9997 \pm 0.0002$ | $0.8537 \pm 0.0747$ | $0.6290 \pm 0.0468$ |

*Table 5.* SLCP $\ell$-C2ST under different simulation budgets. Results are reported as mean $\pm$ standard deviation across the 10 official observations. Lower values indicate better posterior fidelity.

## H. Additional Analysis of FK-Steering

### H.1. Quantitative Analysis on Exoplanet Posterior Estimation

To further evaluate the effect of FK-steering on complex posterior estimation, we compare posterior overlap and mode accuracy on the exoplanet orbital characterization task.

As shown in Table 6, the unsteered model (FUSE w/o FK-Steering) achieves relatively broad posterior coverage but suffers

from substantially worse mode estimation accuracy. By incorporating likelihood-based trajectory correction during sampling, FK-steering improves posterior concentration around the high-likelihood regions and produces posterior samples that more closely match the high-density structure of the reference MCMC solution.

| Method | 95% Credible Region IoU ↑ | Posterior Mode L2 Distance ↓ |
|---|---|---|
| FMPE | 0.33 | 132.41 |
| NPE | 0.29 | 75.74 |
| FUSE w/o FK | 0.52 | 6.60 |
| FUSE | **0.62** | **3.85** |

*Table 6.* Quantitative comparison on the exoplanet orbital characterization task. We report the intersection-over-union (IoU) between the estimated and reference 95% credible regions together with the Euclidean distance between posterior modes. Higher IoU and lower mode distance indicate better agreement with the long-run PTMCMC reference posterior samples. Best results are highlighted in bold and second-best results are underlined.

### H.2. Why FK-Steering Helps on Complex Posteriors

The benefit of FK-steering becomes more pronounced in scientific inverse problems with highly multimodal or degenerate posterior structures. In such settings, standard amortized SBI models may exhibit mass-covering behavior, producing overly broad posterior approximations that fail to accurately capture narrow high-likelihood regions.

FK-steering mitigates this issue by incorporating intermediate likelihood evaluations during the sampling process. By periodically reweighting and correcting the generative trajectories, the method suppresses physically implausible samples and encourages exploration toward regions with higher posterior density.

For relatively simple SBIBM tasks, the base flow-matching model already provides sufficiently accurate posterior estimation, and therefore FK-steering yields only marginal improvements. In contrast, for complex real-world tasks such as exoplanet orbital characterization, FK-steering substantially improves posterior concentration and mode accuracy, as demonstrated in Table 6. Tail coverage remains an empirical property of the finite-particle sampler rather than a formal guarantee.

## I. Theoretical Perspective on FK-Steered Inference

This section provides a theoretical interpretation of the FK-steered sampler used in FUSE. The goal is not to claim that the finite-particle implementation is an asymptotically exact replacement for MCMC. Rather, FK-steering can be viewed as a tractable posterior-tilting correction applied to the learned amortized proposal. This perspective clarifies three design choices: the target path measure, the use of a denoised proxy in simulator-based scoring, and the stochastic rejuvenation used to mitigate particle degeneracy.

### I.1. Posterior-Tilted Path Measure

For a fixed observation $x$, let $\mathbb{Q}_x$ denote the path measure induced by the learned reverse-time sampler conditioned on $x$. A sample path is written as $\theta_{1:0} = \{\theta_t : t \in [1,0]\}$, where $\theta_1$ is initialized from the base distribution and $\theta_0$ is the final generated parameter. Let $q_0^x(\theta)$ be the terminal marginal density of $\mathbb{Q}_x$, and assume $q_0^x(\theta) > 0$ whenever $\gamma_x(\theta) > 0$. If this density were available, an ideal posterior-corrected path measure $\mathbb{P}_x^\star$ could be defined through the Radon–Nikodym derivative

$$\frac{d\mathbb{P}_x^\star}{d\mathbb{Q}_x}(\theta_{1:0}) = \frac{\gamma_x(\theta_0)}{Z_x q_0^x(\theta_0)}, \qquad \gamma_x(\theta) = p(x \mid \theta)p(\theta), \tag{34}$$

where $Z_x$ is the normalizing constant. For any measurable set $A$, the terminal marginal then satisfies

$$\mathbb{P}_x^\star(\theta_0 \in A) = \int_A \frac{\gamma_x(\theta)}{Z_x} \, d\theta = \int_A p(\theta \mid x) \, d\theta. \tag{35}$$

Thus, the ideal change of measure has the exact posterior as its terminal marginal. This construction is a reference density-ratio correction, not the finite-particle algorithm used by FUSE.

In practice, however, the terminal proposal density $q_0^x$ of an implicit flow sampler is unavailable. FUSE therefore uses the unnormalized log posterior as a tractable reward, producing a posterior-tilted refinement of the learned proposal rather than

an exact density-ratio correction. At resampling times $\{t_{r_1}, \ldots, t_{r_J}\}$, the incremental potential is implemented as

$$G_{r_j}(\theta_{t_{r_j}}) = \exp\left(\frac{\lambda}{J} r_\phi(\theta_{t_{r_j}}, t_{r_j}; x)\right), \tag{36}$$

where $r_\phi$ is defined in Eq. 11. The resulting particle system is a bootstrap approximation of this Feynman–Kac flow: particles are propagated by the learned sampler and resampled according to the normalized scores $G_{r_j}^k / \sum_{\ell=1}^{B} G_{r_j}^\ell$. Because the implementation uses the learned reverse transition as the proposal, the transition correction in the general FK importance weight is one; if a different proposal were used, the corresponding transition-density ratio would have to be included. Even with infinitely many particles, these chosen potentials target the associated surrogate FK-tilted path measure rather than the ideal density-ratio-corrected posterior path measure above; finite $B$ adds the usual SMC approximation error.

## I.2. Why Scoring the Denoised Proxy is Principled

A key design choice is to evaluate the simulator likelihood at the denoised proxy $\hat{\theta}_t$, rather than at the noisy intermediate state $\theta_t$. This is important because $\theta_t$ is a mixture of clean parameters and noise and may not lie on the physical parameter manifold. Feeding such noisy states into a scientific simulator can violate parameter constraints and produce meaningless likelihood values.

Under the rectified-flow interpolation used in training, $\theta_t = (1-t)\theta_0 + t\epsilon$, with velocity target $\epsilon - \theta_0$. In the idealized case where $v_\phi$ equals the optimal conditional velocity, rearranging the interpolation gives

$$\mathbb{E}[\theta_0 \mid \theta_t, x] = \theta_t - t\, v^\star(\theta_t, t, x), \tag{37}$$

which motivates the plug-in proxy $\hat{\theta}_t = \theta_t - t\, v_\phi(\theta_t, t, x)$ used in Eq. 11. The ideal intermediate FK lookahead potential would be

$$U_t(\theta_t) = \log \mathbb{E}_{\mathbb{Q}_x}\left[\gamma_x(\theta_0) \mid \theta_t\right], \tag{38}$$

which is generally intractable because it integrates over all terminal states reachable from $\theta_t$. A local Taylor expansion around the denoised conditional mean yields the plug-in approximation

$$U_t(\theta_t) \approx \log p(x \mid \hat{\theta}_t) + \log p(\hat{\theta}_t). \tag{39}$$

This derivation explains why the simulator-based reward in FUSE is evaluated at $\hat{\theta}_t$. The approximation is most accurate near the clean data manifold and should be interpreted as a tractable likelihood-guided correction rather than an exact posterior sampler.

## I.3. SDE Rejuvenation and Under-Dispersion

Resampling concentrates computation on high-reward trajectories, but repeated resampling can reduce particle diversity. FUSE mitigates this by using a stochastic sampler after resampling. The role of the SDE perturbation is to restore local exploration while preserving the learned probability path as much as possible.

The stochastic term in Eq. 10 is controlled by $\sigma_t = \alpha\sqrt{t/(1-t)}$. In practice the sampler avoids the endpoints and clips the denominator for numerical stability. Larger $\alpha$ increases exploration but may weaken likelihood concentration; smaller $\alpha$ makes the sampler closer to deterministic FK selection and may increase under-dispersion. Thus, particle count, resampling frequency, and noise scale jointly control the fidelity-diversity trade-off. Empirically, this trajectory-level correction is more effective than deterministic post-hoc selection on the $\beta$ Pictoris b task, as shown in Appendices H.1 and F.

In summary, FK-steering should be viewed as a likelihood-guided correction to an amortized posterior sampler. Its ideal path-measure formulation connects to posterior tilting, while the implemented algorithm uses a denoised plug-in reward and finite-particle SMC approximation. This provides a practical mechanism for improving high-density posterior fidelity without sacrificing the speed advantage of amortized inference.

