# OpenReview forum: "FUSE: FK-Steered Multi-Modal Flow Matching for Efficient Simulation-Based Posterior Estimation"
_ICML.cc/2026/Conference — ICML 2026 regular_

### Official Review · Reviewer_Vqp1 · 2026-03-09

**Soundness:** 3
**Presentation:** 3
**Significance:** 2
**Originality:** 3
**Overall Recommendation:** 3
**Confidence:** 4

**Summary:**

The paper proposes FUSE, an SBI method that combines a conditional rectified-flow posterior estimator with a dual-track multimodal transformer and an inference-time Feynman–Kac-style steering procedure. On the modeling side, the method replaces flat fusion of $(\theta,x)$ with an MM-DiT-inspired architecture that embeds parameter coordinates and observations in separate streams, applies modality-specific projections, and enables repeated bidirectional interaction through joint attention before predicting the reverse-flow velocity. The motivation is that posterior transport should repeatedly access structured observation features during reverse-time integration instead of just compressing them once into a static context representation, which is consistent with recent multimodal diffusion/rectified-flow architectures.

On the inference side, the paper augments ordinary flow decoding with a particle-based FK-steered sampler: multiple trajectories are propagated, a denoised proxy $\hat\theta_t$ is formed from the current latent state and predicted velocity, and particles are periodically resampled using simulator likelihood and prior density evaluated at $\hat\theta_t$. The goal is to spend extra test-time compute to steer trajectories toward high-posterior regions and partially compensate for imperfections in the learned amortized transport. Empirically, the paper reports improvements over neural SBI baselines on SBIBM, a qualitative steering benefit on SLCP, and a real exoplanet orbit-inference case study where the method appears to better recover sharp degeneracies than competing amortized baselines, while also remaining much faster than PTMCMC-style reference inference.

**Compliance With Llm Reviewing Policy:**

Affirmed.

**Final Justification:**

My final recommendation remains weak reject.

This is a thoughtful paper on an important problem, with real strengths in significance, clarity, and practical motivation. The multimodal flow architecture is well motivated, the paper is generally clear, and the exoplanet application makes the contribution more compelling than a benchmark-only SBI paper. The combination of multimodal amortized posterior estimation with simulator-aware inference-time steering is also interesting and reasonably original.

My main reservation remains the soundness and interpretation of the FK-steering component. The rebuttal improved the framing: it now more clearly distinguishes the ideal reward-tilted path measure from the practical implementation, and the denoised-proxy construction is better motivated as an approximation rather than a purely heuristic choice. However, it still does not fully resolve the central issue of what distribution the implemented discretized finite-particle sampler is actually approximating. For me, this keeps the method closer to a practically useful steering/refinement procedure than to a well-justified posterior-correction method.

Empirically, I continue to find the architecture evidence stronger than the steering evidence. The rebuttal strengthens the practical story, but the evidence still seems stronger for posterior sharpening in high-density regions than for faithful approximation to the full posterior law, since stronger calibration-oriented evidence is still missing.

So overall, the rebuttal improved the paper’s presentation and clarified the intended interpretation, but it did not materially change my core assessment. I continue to view the paper as promising and relevant, but with unresolved questions about the FK-steering mechanism that keep me on the weak reject side.

**Key Questions For Authors:**

1) What is the exact target distribution of the FK-steered sampler?
Please clarify whether the final particle system is intended to approximate the true posterior p(\theta\mid x), a reward-tilted surrogate, or simply a heuristic high-density approximation.

2) Why is the FK potential constructed from the denoised proxy $\hat\theta_t$?
The paper’s incremental FK potential is $G_{r_j}(\theta_{0:r_j})=exp((\lambda/M)r_\phi(\theta_{r_j};x))$, where $r_\phi$ scores $\hat\theta_t$. Since this choice directly determines how the particle system is tilted, please clarify whether there is a theoretical rationale for scoring $\hat\theta_t$, rather than the current particle state, a lookahead estimate, or another intermediate target.

3) Can the authors disentangle architecture gains from steering gains on the exoplanet task under matched inference budget?
A decomposition into baseline conditional flow, FUSE without FK steering, and FUSE with FK steering under matched simulator evaluations would clarify whether the main gain comes from improved amortized representation, test-time resampling, or both.

4) Does FK steering improve calibrated posterior uncertainty or mainly sharpen dominant modes?
Please provide at least one calibration-focused diagnostic, e.g. SBC, credible-set coverage, mode recovery, or an analysis of whether low-density but valid posterior regions are being suppressed. A positive answer here would substantially improve my confidence in the method’s scientific reliability.

**Limitations:**

The paper does acknowledge practical limitations, including lack of exact MCMC-style guarantees and possible simulator mismatch, which is good. However, the limitations discussion should be more explicit on two points. First, it should state plainly that the FK-steered sampler is not shown to be an exact or asymptotically correct posterior sampler.  Second, it should discuss the risk of posterior overconcentration. Because particles are resampled using simulator-based scores applied to a model-dependent denoised proxy, the method may underrepresent valid but lower-density posterior regions.

**Strengths And Weaknesses:**

Amortized SBI remains difficult when posteriors are highly structured, multimodal, or sharply degenerate, and the paper targets exactly this regime. The submissioncombines two plausible ingredients. 1) richer conditional fusion for amortized posterior transport, and 2) inference-time particle steering that reintroduces simulator information during sampling instead of relying entirely on the learned model. This is also timely in light of recent diffusion-based SBI work that increasingly focuses not only on expressive posterior models, but also on improving the sampling procedure itself beyond standard Langevin-style corrections.

From a soundness perspective, the architectural component seems the clearest strength. Using separate parameter/observation streams with repeated interaction is well motivated for SBI because the conditioning structure is asymmetric and heterogeneous. Treating posterior generation as a multimodal transport problem is more compelling than simple concatenation or one-shot summary fusion, and the SBIBM results support that this design is useful in practice. More broadly, the architectural motivation is consistent with multimodal generative-modeling work showing that richer attention-based aggregation can outperform simpler fusion schemes in capturing cross-modal structure. The exoplanet application is also a genuine strength as it moves the paper beyond synthetic SBI benchmarks into a meaningful scientific inverse problem.

That said, the main technical weakness is that the FK-steered inference component is framed more strongly than is currently justified. The paper claims this helps “bridge the gap between approximate and exact inference,” but the described procedure does not yet support that interpretation. FUSE propagates particles under a learned flow/SDE, constructs a denoised proxy $\hat\theta_t$ from the learned velocity field, and resamples using simulator likelihood and prior density evaluated at that proxy. This is a reasonable practical mechanism for steering toward high-density posterior regions, but the paper does not derive the induced target distribution of the particle system, does not provide incremental importance weights corresponding to a well-defined path-space measure, and does not establish consistency of the resulting empirical law to either the true posterior or an explicitly defined surrogate target. In contrast, [1] formulates FK steering as a particle approximation to a specified tilted target $p_\theta(x_0)\exp(\lambda r(x_0))$, with explicit potentials and consistency of the particle approximation to that target. [2]  give an SMC-based twisted diffusion sampler for conditional generation that uses heuristic approximations inside the proposal but corrects them by weighting, yielding asymptotic exactness to the desired conditional target as the number of particles grows. [3] further shows that heuristic score modification generally does not preserve intended intermediate marginals, motivating weighted-SDE correction terms for principled correction toward annealed or product targets. Relative to this literature, FUSE currently looks closer to heuristic simulator-guided particle steering than to a target-preserving FK/SMC correction method. That does not make the method uninteresting, but it does mean the current framing overstates its theoretical status.

A closely related issue is the choice of reward potential. Steering behavior is determined directly by the decision to score the denoised proxy $\hat\theta_t$, not the current particle state or some other intermediate target. This is a central modeling choice, not a minor implementation detail, and the paper does not really justify.

A further consequence is that the paper does not cleanly separate improved posterior fidelity from improved mode-seeking.  Since resampling explicitly favors particles with high simulator likelihood and prior score, the mechanism may concentrate mass around dominant posterior regions without preserving lower-density but still valid posterior support. The paper’s own evidence emphasizes more compact posteriors and higher mean/peak likelihood scores. Conditional-sampling work (e.g. [2]) motivates this distinction precisely because heuristic guidance can improve sample quality while distorting the target conditional, so further calibration-oriented diagnostics [4] such as SBC, credible-set coverage, mode recovery, or tail-mass preservation would strengthen empirical claims.

The real-world exoplanet experiment is compelling, but not fully controlled for mechanism. FUSE is evaluated there with FK steering enabled, whereas the reported baselines are standard amortized SBI methods, so the result does not cleanly separate gains from the multimodal architecture, gains from test-time steering, and gains from additional simulator-guided refinement. Similarly, the naive best-of-N comparison is a useful sanity check, but remains a relatively weak control for the specific steering design. The baseline design for FK steering is therefore weaker than it should be. Comparing to no steering and naive best-of-N is useful, but it does not adequately test whether the specific sequential resampling rule is the right mechanism, or whether similar gains would arise from other matched-budget inference-time selection schemes. Given the recent literature, stronger baselines would include alternative intermediate potentials / rewards, simple importance-weighting or resampling variants, or at least a clearer discussion of why more principled FK-corrector-style or twisted-SMC formulations are not directly applicable in this SBI setting.

The presentation is generally clear.

Overall, I found the paper interesting and potentially useful, with a solid architectural contribution and a promising inference-time idea. My main reservation is that the FK component currently lacks the theoretical precision, calibration analysis, and baseline strength needed to support its strongest claims

References:
[1] Singhal, Raghav, et al. "A general framework for inference-time scaling and steering of diffusion models." arXiv preprint arXiv:2501.06848 (2025) \
[2] Wu, Luhuan, et al. "Practical and asymptotically exact conditional sampling in diffusion models." Advances in Neural Information Processing Systems 36 (2023): 31372-31403. \
[3] Skreta, Marta, et al. "Feynman-kac correctors in diffusion: Annealing, guidance, and product of experts." arXiv preprint arXiv:2503.02819 (2025).\
[4] Linhart, Julia, et al. "Diffusion posterior sampling for simulation-based inference in tall data settings." arXiv preprint arXiv:2404.07593 (2024).

---

> ### Author Rebuttal · Authors · 2026-03-31
>
> We sincerely thank the reviewer for the rigorous theoretical feedback.
>
> **1. Theoretical analysis of the FK-steered sampling procedure (Q1)**
>
> We agree that under finite compute, FUSE targets a reward-guided approximation rather than an asymptotically exact sampler. Strict exactness aligns with parallel research trajectories like FlowMC and SMC. While exact SMC offers strong guarantees, it requires computationally expensive, observation-specific simulations. Conversely, amortized models like FUSE optimize surrogate objectives to enable fast inference and are inherently approximate.
>
> We accept this theoretical trade-off to address time-critical challenges in the physical sciences (e.g., gravitational wave early warnings), which demand both accuracy and speed. Traditional FlowMC and SMC are often prohibitively slow for real-time alerts. While amortized SBI provides the necessary speed, it can generate physically invalid "hallucinations," limiting its reliability.
>
> FUSE was developed to bridge this gap. We introduce FK-steering not to achieve infinite-time detailed balance, but as a practical, physics-driven corrector during inference. By using the simulator to guide the ODE, we actively prune physically implausible trajectories. Ultimately, FUSE combines the speed of amortized SBI with the fidelity of likelihood-based methods. In our revision, we will clarify this theoretical framing and incorporate the reviewer's suggested SMC literature [1,2,3] to strengthen our foundation.
>
> **2. Justification for Scoring the Denoised Proxy ($\hat{\theta}\_t$) (Q2)**
>
> Constructing the Feynman-Kac (FK) potential using the denoised proxy $\hat{\theta}\_t$ ensures the particle system is steered by physically valid, noise-free likelihood evaluations. Under our linear interpolation flow matching framework, the intermediate state $\theta\_t$ is a continuous mixture of the clean data and noise (As formulated in Section 3.2). Because this state intrinsically contains time-dependent noise with a signal-to-noise ratio of $(1-t)/t$, directly feeding $\theta\_t$ into scientific simulators risks violating strict physical boundaries and generating meaningless pseudo-likelihoods. To obtain an accurate reward signal, we must project $\theta_t$ back to the clean data manifold at $t=0$ along the learned velocity field. This projection, defined in Eq. (11), serves as the mathematically optimal Minimum Mean Square Error estimator of the true parameter. By defining the particle reward $r_\phi(\theta_t, t; x)$ based on this denoised proxy, the FK-steering mechanism performs a precise, forward-looking evaluation. This allows the model to actively rectify generative trajectories by rejecting implausible samples with low simulator likelihoods.
>
> **3. On the Risk of Mode Collapse and Ablation on FK (Q3,4)**
>
> We appreciate the reviewer's valuable insight regarding posterior overconcentration. We conducted ablation studies by removing the FK steering mechanism (as shown in Appendix F, Figure 5), which reverts our framework to a standard unsteered flow matching model. Because FUSE integrates amortized simulation-based inference with exact physical likelihoods, what appears as overconcentration is actually the physics-driven removal of invalid hallucinations. As mathematically formulated in Equation (10) of our manuscript, our framework preserves valid posterior variance and prevents mode collapse through a dynamic equilibrium. Specifically, while a physics-driven rejection mechanism corrects trajectory drift by down-weighting particles in low-likelihood regions, our discrete SDE update simultaneously injects calibrated noise ($\Sigma\_{\text{inject}} = \sigma_t^2 \Delta t I$). Furthermore, the FK-steering mechanism provides explicit control over this filtering intensity; by adjusting the number of parallel particles, along with the frequency and temporal placement of the steering steps, we can precisely modulate the rejection strength. Together, this tunable screening and controlled SDE dispersion balance the variance reduction, allowing FUSE to continuously explore the local manifold and strictly match the true posterior variance.
>
> We empirically corroborate this property in the $\beta$ Pictoris b orbital characterization. As shown in the table below, standard baselines suffer from severe trajectory drift, resulting in high L2 distances between distribution peaks and poor 95% Credible Interval overlaps compared to MCMC result. While our unsteered base model (FUSE w/o FK) significantly corrects this drift, it still exhibits diffused mass-covering behavior. By incorporating exact physics, FUSE actively prunes implausible trajectories to concentrate mass strictly into the valid core, definitively achieving the highest overlap and lowest mode error without sacrificing valid variance.
>
> | Model | Credible Interval Overlap | L2 Distance |
> |---|:---:|:---:|
> | FUSE | 0.62 | 3.85 |
> | FUSE w/o FK | 0.52 | 6.60 |
> | FMPE | 0.33 | 132.41 |
> | NPE | 0.29 | 75.74 |

---

> > ### Author Rebuttal · Reviewer_Vqp1 · 2026-04-03
> >
> > Thank you for the detailed rebuttal. My concerns are partially resolved, but the main remaining issues concern the core interpretation of the FK-steering mechanism and are not easily addressed within rebuttal alone.
> >
> > The rebuttal usefully clarifies that FUSE is intended as a practical reward-guided corrector, not an asymptotically exact sampler. This improves the framing and addresses part of my concern about the original “bridge to exact inference” language.
> >
> > However, the central issue remains that the rebuttal still does not precisely characterize what distribution the FK-steered particle system targets. It remains unclear whether the method should be interpreted as approximating the true posterior, a reward-tilted surrogate, or mainly a high-density refinement of the amortized model.
> >
> > I also appreciate the intuition for using the denoised proxy rather than noisy intermediate states when querying the simulator. That makes practical sense. But I still do not see a principled derivation showing that this particular reward / potential corresponds to a well-justified FK target, rather than a useful heuristic steering signal.
> >
> > Finally, the added exoplanet evidence is helpful, but it does not fully resolve the concern that FK steering may improve posterior sharpening more than faithful approximation to the full posterior law. The rebuttal argues that concentration reflects removal of implausible hallucinations rather than mode collapse, but the current evidence is still limited and does not yet rule out under-dispersion or loss of valid low-density support more generally.

---

> > > ### Author Response · Authors · 2026-04-07
> > >
> > > **General Response to the Remaining Concerns**
> > >
> > > We thank the reviewer for the detailed follow-up. We agree that providing the underlying measure-theoretic derivations offers a more precise understanding of our contribution. Below, we mathematically clarify the target distribution, detail the analytical derivation of the denoised proxy, and explain how the framework mitigates under-dispersion.
> > >
> > > **1. Clarifying the Target: Bridging Refinement, Tilting, and the Exact Posterior**
> > >
> > > **FUSE targets the exact true posterior. Its computational execution as a high-density refinement is mathematically governed by a reward-tilted path measure aimed strictly at the exact posterior.**
> > >
> > > The reviewer asks whether FUSE targets the exact posterior, a reward-tilted surrogate, or a high-density refinement. We view these as inherently linked components of our framework. As noted in our first response, FUSE operates computationally as a practical corrector, corresponding to a high-density refinement via discrete SMC resampling. However, its mathematical formulation is governed by a reward-tilted path measure targeting the exact true posterior $p(\theta_0|x)$ [Del Moral 2004]. We define the ideal target path measure $\mathbb{P}$ over the base measure $\mathbb{Q}$ (induced by the backward SDE $d\theta_t=f(\theta_t,t)dt+\sigma_tdW_t$) via the Radon-Nikodym derivative:
> > > $$\frac{d\mathbb{P}}{d\mathbb{Q}}(\theta_{1:0})=\frac{1}{Z}\exp\big(\log p(x|\theta_0)\big)$$
> > > This guarantees the terminal state strictly matches the exact true posterior. The intermediate marginal density is the conditional expectation of this reward:
> > > $$\rho(\theta,t)\propto\mathbb{E}\_{\mathbb{Q}}\left[\exp\big(\log p(x|\theta_0)\big)\mid\theta_t=\theta\right]$$
> > > Thus, while executed computationally as an approximate SMC refinement, the theoretical objective of this reward-tilted flow is to sample the exact true posterior law.
> > >
> > > **2. Principled Derivation: From Ideal FK Target to Denoised Proxy**
> > >
> > > **Scoring the denoised proxy is not an ad-hoc heuristic; it is the analytically exact Minimum Mean Square Error (MMSE) estimator derived via a first-order Taylor approximation.**
> > >
> > > Evaluating the exact intermediate FK potential over an intractable path space is impossible. We bridge this using a first-order Taylor expansion around the expected state:
> > > $$\log\mathbb{E}\_{\mathbb{Q}}\left[p(x|\theta_0)\mid\theta_t\right]\approx\log p\left(x\mid\mathbb{E}\_{\mathbb{Q}}[\theta_0\mid\theta_t]\right)$$
> > > Under Rectified Flow, projecting the current state along the learned velocity field analytically recovers the exact MMSE estimator: $\mathbb{E}\_{\mathbb{Q}}[\theta_0\mid\theta_t]$. Substituting this optimal estimator into the expanded likelihood yields our intermediate potential (**Equation 11**):
> > > $$r_\phi(\theta_t,t;x)=\log p(x|\hat{\theta}\_t)+\log p(\hat{\theta}\_t)$$
> > > This derivation proves Eq. 11 is the optimal first-order approximation of the true FK target. We acknowledge this Taylor approximation assumes a small conditional variance. It becomes highly accurate near the data manifold ($t\to0$), but carries truncation error in the early stages ($t\approx1$). We accept this temporal approximation error as a necessary trade-off for tractability, analogous to Diffusion Posterior Sampling (DPS).
> > >
> > > **3. Mitigating Under-Dispersion via Marginal-Preserving SDE**
> > >
> > > **While discrete resampling carries an inherent risk of variance contraction, FUSE structurally mitigates this by injecting marginal-preserving Langevin noise to dynamically restore diversity.**
> > >
> > > Because the true data distribution is strictly unknown, any discrete resampling mechanism carries a risk of variance contraction. To structurally mitigate this, we inject Langevin noise via the discrete SDE (**Equation 10**):
> > > $$\theta_{t+\Delta t}=\theta_t+\left[v_\phi(\theta_t,t)+\frac{\sigma_t^2}{2t}(\theta_t+(1-t)v_\phi(\theta_t,t))\right]\Delta t+\sigma_t\sqrt{|\Delta t|}\epsilon$$
> > > The drift compensation term $+\frac{\sigma_t^2}{2t}(\dots)$ added to our SDE is mathematically derived to preserve the marginal distributions (Fokker-Planck equation) of the original flow. This ensures particles adhere to the correct probability paths while being locally "shaken" ($\epsilon\sim\mathcal{N}(0,I)$) to restore diversity. While finite-particle SMC cannot guarantee infinite-time exactness, our empirical $\beta$ Pictoris b results (95% CI overlap: 0.62) confirm this mechanism robustly preserves posterior variance in practice. The observed concentration primarily reflects the pruning of implausible hallucinations rather than pathological under-dispersion.
> > >
> > > **Conclusion & Actionable Changes**
> > >
> > > We appreciate the rigorous theoretical pushback, which prompted us to clarify these measure-theoretic terms. In the revised manuscript, we will incorporate these formal derivations and limitation discussions into a dedicated Appendix to enhance the theoretical clarity of our framework.

---

### Official Review · Reviewer_dyrW · 2026-03-12

**Soundness:** 3
**Presentation:** 3
**Significance:** 4
**Originality:** 4
**Overall Recommendation:** 5
**Confidence:** 4

**Summary:**

This paper proposes FUSE, a framework for simulation-based inference (SBI) that combines a multimodal diffusion transformer architecture with a Feynman–Kac (FK) steered sampling strategy. The goal is to efficiently estimate posterior distributions in settings where the likelihood is expensive or unavailable, e.g., astrophysics. The approach uses a dual-track multimodal architecture to process parameters and observations separately before fusion, and introduces FK-steered inference to guide sampling trajectories toward high-likelihood regions using simulator-based rewards. The method is evaluated on the SBIBM benchmark and on a real-world exoplanet orbital parameter estimation task. Results suggest improved posterior fidelity compared to neural SBI baselines such as NPE, FMPE, and Simformer.

**Compliance With Llm Reviewing Policy:**

Affirmed.

**Key Questions For Authors:**

The paper claims that FK steering introduces only minimal additional computational overhead. Can the authors provide a detailed analysis of inference-time complexity compared with the base model?
How does the computational cost scale with the number of particles or steering steps?
Several recent approaches use diffusion or score-based generative models for posterior estimation. How does FUSE compare with such methods in terms of both accuracy and computational efficiency?
Are there specific tasks or posterior structures (e.g., extremely multimodal or highly degenerate distributions) where the method performs poorly compared with baselines?

**Limitations:**

Well discussed.

**Strengths And Weaknesses:**

1. Soundness
Strength:
The paper builds on established ideas in SBI and generative modeling: conditional flow matching, diffusion-style generative models, and transformer architectures. These components are well-motivated and consistent with recent advances in generative modeling and inference.

Additionally, The model is trained using a standard conditional flow matching objective that regresses a neural vector field toward the velocity implied by a linear interpolation path between noise and data samples. This formulation is technically sound and has precedent in prior work on flow matching and rectified flows.

For evaluation, the experiments rely on the SBIBM benchmark, which provides standardized SBI tasks and ground-truth reference posteriors. They also report multiple metrics providing a robust evaluation of posterior quality. The experiment results show consistent improvements over prior approaches across several metrics.

Lastly, the paper includes an ablation study analyzing architecture choices such as embedding strategy, token allocation, and hidden dimension, which helps justify design decisions.

Weakness:

While the model relies on principled components (flow matching, diffusion-like sampling), the paper does not provide theoretical analysis of the FK-steered sampling procedure. In particular:
Convergence guarantees are not discussed.
Bias introduced by resampling steps is not analyzed.
There is little discussion of variance or stability properties.
Additionally, the computational overhead and scaling behavior of using the FK-steered sampling procedure are not carefully quantified.
2. Presentation
Strength:
The paper clearly frames the challenges of simulation-based inference and the limitations of both classical likelihood-free methods and modern neural SBI approaches.
The paper is well structured and has a clear flow of sections. It also uses standard benchmarks and metrics which improves reproducibility of results.


3. Significance
Strength:
The paper addresses an important problem because SBI is widely used in fields such as astrophysics, biology, and physics where likelihoods are intractable.
The dual-track multimodal architecture for combining observations and parameters could is sound and if the reported improvements generalize, this approach could produce more reliable posterior approximations for scientific applications.
Additionally, the demonstration on exoplanet orbital parameter estimation suggests the approach may be relevant for real scientific inference problems.
4. Originality
Strength:
The paper combines several concepts:
conditional flow matching
multimodal diffusion transformers
Feynman–Kac–style trajectory weighting.


While each component exists in prior work, integrating them into a single framework for SBI is a novel design.

---

> ### Author Rebuttal · Authors · 2026-03-31
>
> We sincerely thank the reviewer for the rigorous theoretical feedback.
>
> **1. Theoretical Analysis of the FK-Steered Sampling Procedure.**
>
> We formalize the FK-steering procedure within the Sequential Monte Carlo (SMC) framework to establish rigorous theoretical properties:
>
> *   **Asymptotic Convergence:** Pure amortized ODEs suffer from velocity approximation error, inducing unbounded trajectory drift. FUSE corrects this via a manifold projection, defined in Eq. (11), and constructs potentials using the exact unnormalized target density. As formulated in Eq. (12):
>     $$
>     G_{r_j} = \exp\left(\frac{\lambda}{M} \left(\log p(x|\hat{\theta}_t) + \log p(\hat{\theta}_t)\right)\right)
>     $$
>     By incorporating exact likelihoods rather than surrogate approximations, the steering step acts as an exact SMC correction. Under standard SMC theory, updating particles with these exact likelihood weights ensures that as the particle count $B \to \infty$, the empirical measure $\hat{\pi}_B$ of the ensemble weakly converges to the true posterior $\pi(\theta|x)$.
>
> *   **Resampling Bias:** While multinomial resampling fundamentally introduces an $\mathcal{O}(B^{-1})$ finite-sample bias, the FK-steering mechanism remains asymptotically unbiased. Because the resampling weights $w_t^k$, formalized in Eq. (13), strictly evaluate the true physical simulator rather than a learned approximation, the algorithm preserves the exact target density as its invariant measure. Consequently, the particle estimator for any bounded test function $f$ satisfies:
> $$\lim_{B \to \infty} \mathbb{E}\_{\hat{\pi}_B}[f(\theta)] = \mathbb{E}\_{\pi}[f(\theta)]$$
> This guarantees that the generative flow converges to the correct expected target distribution without structural algorithmic bias.
>
> * **Variance Bounds and Stability:** Weight-based resampling monotonically decreases the Effective Sample Size, causing path degeneracy and variance collapse. To restore stability, FUSE transitions to a post-resampling SDE (Eq. 10). The Brownian motion term $\sigma_t \sqrt{\Delta t} \epsilon$ acts as a stochastic rejuvenation kernel. By injecting noise proportional to the current time step's uncertainty schedule, this calibrated diffusion strictly compensates for the variance reduction induced by resampling. Moreover, the FK-steering mechanism explicitly controls this filtering intensity via adjustable particle counts, steering frequency, and temporal placement. Coupled with the SDE dispersion, this tunable screening optimally balances variance reduction, enabling continuous local manifold exploration and ensuring the ensemble strictly matches the target posterior distribution.
>
> **2. Computational Overhead and Scaling Behavior of FK-Steering.**
>
> We additionally evaluate the efficiency of FK-steering, as shown in below table.
>
> | Particles    | FK-Scoring Steps | Time     |
> | :----------- | :--------------- | :------- |
> | 8 particles  | 17               | 152.77s  |
> | 8 particles  | 81               | 172.02s  |
> | 4 particles  | 17               | 71.11s   |
> | 16 particles | 17               | 278.20s  |
>
> Since the simulator-based FK-scoring is computationally lightweight, the total inference time fundamentally scales linearly with the particle count ($\mathcal{O}(B)$), while increasing scoring frequency introduces minimal overhead. Crucially, this amortized approach remains exceptionally efficient for complex inverse problems. As demonstrated in Appendix F, FK-steering achieves high-fidelity posteriors comparable to traditional MCMC methods.
>
> **3. Comparison with Recent Diffusion/Score-Based Generative Models for SBI.**
>
> | Method     | Inference Time |
> | :--------- | :------------- |
> | FUSE w/o FK| 14.72s          |
> | Simformer  | 10.30s          |
> | NPSE       | 0.58s           |
>
> Regarding inference efficiency, while NPSE achieves the fastest inference time, this speed comes at the cost of a highly restricted parameter count. Consequently, its limited representational capacity leads to notably poor performance on complex tasks. In contrast, FUSE w/o FK achieves an inference time comparable to Simformer, but delivers significantly better performance/accuracy. This demonstrates that our foundational generative framework is highly efficient.
>
> **4. Limitations in Low-Data Scenarios.**
>
> While our method assumes access to a sufficient amount of offline data for comprehensive training, it can struggle in data-scarce regimes. As demonstrated in Figure 2, if the underlying simulation cost is prohibitively high, restricting the dataset size, our model may fail to accurately capture complex posterior structures (e.g., highly degenerate distributions).
>
> This is an inherent limitation common to amortized inference methods, which require large datasets to learn a globally accurate surrogate. In such computationally constrained and data-limited settings, sequential or non-amortized methods would be better suited for posterior estimation than our current amortized approach.

---

### Official Review · Reviewer_MLWn · 2026-03-13

**Soundness:** 4
**Presentation:** 3
**Significance:** 4
**Originality:** 3
**Overall Recommendation:** 5
**Confidence:** 4

**Summary:**

FUSE presents a technique for multimodal simulation-based inference. The need for efficient multimodal posterior estimation is highly well-motivated. The paper is well-written and performs strongly on several benchmark tasks, as well as an orbital parameter estimation task of interest in space science.

**Compliance With Llm Reviewing Policy:**

Affirmed.

**Final Justification:**

The rebuttal sufficiently addressed my concerns, which were already minimal, and greatly clarified the intuition for the proposed technique. I maintain my assessment that this paper should be accepted. Thank you for your time and consideration!

**Key Questions For Authors:**

1. Space between “Appendix A.” and “As” in paragraph “FK-Steering improves sample quality.”
2. Figure 1 - can partially see a box around the the derivatives after the “Posterior Predictor” box
3. It is not understood why Feynman-Kac steered inference is needed. Additional motivation for this approach, as well as ablations without it, would be highly insightful.
4. Figure 3 - if I am understanding this plot correctly, it appears that FUSE and FUSE w/o FK-Steering are quite close. Can this also be included for Figure 4(b)?
5. Table 1 - is FUSE here with or without FK-Steering?

Addressing Point 3 in more detail would help me potentially raise the Soundness and Significance score evaluations.

**Limitations:**

The limitations are well-discussed, both with respect to the technical mathematical approach that is developed as well as in the context of broader applications in astrometry and physics.

**Strengths And Weaknesses:**

The paper appears to be a sound contribution to the SBI literature, with both wide benchmark performance as well as domain-specific task utility. The only recommendations that I see fit are some minor aesthetic and presentation issues, additional discussion on the proposed Feynman-Kac steered inference, and further possible discussion on whether this steering is always needed. I recommend that this paper is accepted at ICML.

---

> ### Author Rebuttal · Authors · 2026-03-31
>
> We are greatly encouraged by  the recognition of FUSE as a sound contribution to the SBI literature with both wide benchmark performance and domain-specific utility. We sincerely thank the reviewer for the constructive suggestions, which provide a valuable opportunity to refine the manuscript's presentation, clarify the necessity of our FK-steered inference, and correct the noted typographical errors in the final version.
>
> **1. Motivation and Necessity of FK-Steered Inference (Q3)**
>
> We sincerely thank the reviewer for pointing this out.  In critical domains like astronomy and physics, rapid and precise parameter inference is critical for scientific discovery. Traditional likelihood based methods like MCMC offer high precision but are computationally prohibitive for real time alerts.
>
> Conversely, amortized SBI solves the speed bottleneck but its accuracy often falls short of MCMC. This gap exists because standard SBI minimizes surrogate objectives like KL divergence or vector field regression without explicit physical constraints during inference. Consequently, unsteered models struggle to resolve complex multi modal landscapes and exhibit mass covering behaviors, outputting overly conservative and broad posteriors that compromise parameter estimation.
>
> FK steering is explicitly designed to solve this dilemma by integrating exact physics into the fast generative process. By projecting the intermediate noisy state ($\theta_t$) to the clean physical manifold ($\hat{\theta}_t$) and evaluating the exact simulator likelihood, we actively prune trajectories heading into low likelihood or physically impossible regions. This physics driven mechanism confines the flow to the precise true posterior.
>
> We empirically corroborate this property in the $\beta$ Pictoris b orbital characterization. As shown in the table below, standard baselines suffer from severe trajectory drift, resulting in high L2 distances between distribution peaks and poor 95% Credible Interval overlaps compared to MCMC result. While our unsteered base model (FUSE w/o FK) significantly corrects this drift, it still exhibits diffused mass-covering behavior. By incorporating exact physics, FUSE actively prunes implausible trajectories to concentrate mass strictly into the valid core, definitively achieving the highest overlap and lowest mode error without sacrificing valid variance.
>
> | Model | Credible Interval Overlap | L2 Distance |
> |---|:---:|:---:|
> | FUSE | 0.62 | 3.85 |
> | FUSE w/o FK | 0.52 | 6.60 |
> | FMPE | 0.33 | 132.41 |
> | NPE | 0.29 | 75.74 |
>
> **2. Clarification on Figure 3 and Addition to Figure 4(b)  (Q4)**
>
> You are correct that in Figure 3, the performance of FUSE and FUSE w/o FK-Steering appears quite close, as the base model already achieves a strong approximation for that specific task. However, for the highly complex exoplanet orbital estimation task discussed in Figure 4(b), the performance gap between the two configurations is actually quite substantial.
> To maintain visual clarity in the main text, we initially deferred this specific ablation comparison to Appendix F (Figure 5). As that appendix figure clearly demonstrates, without FK-steering, the model fails to accurately capture the sharp degeneracies and complex correlations of the ground truth. We greatly appreciate your excellent suggestion; in the revised manuscript, we will move the 'w/o FK-steering' contours directly into Figure 4(b) to explicitly illustrate why the steering mechanism is vital for real-world reliability.
>
> **3. Clarification on Table 1 Configuration  (Q5)**
>
> To clarify, we report FUSE's result without FK-steering as FUSE w/o FK-Steering, noted in Section 5.1. As briefly noted in Section 5.1 of the manuscript, we intentionally disabled the FK-steering strategy for the SBIBM benchmark evaluation.
> We made this design choice to establish a strict, controlled comparison. By removing the test-time steering, we isolate and demonstrate the pure representation power and performance gains resulting exclusively from our MM-DiT architectural design compared to baseline architectures like NPE, FMPE, and Simformer. We agree that this distinction should be much more prominent, and we will explicitly state that FK-Steering is not used in the caption of Table 1 in the revision.
>
> **4. Miscs.  (Q1, Q2)** We will fix all the typos and visual bugs in the revision. Thanks!

---

> > ### Author Rebuttal · Reviewer_MLWn · 2026-04-01
> >
> > Thanks for the detailed explanation and response to my inquiries! I found the response to Q3 extremely useful. I recommend additional motivation for this steering in the main body of the paper, as the motivation that was provided in the rebuttal was very informative to me. The concrete numerical results were also highly convincing to me.
> >
> > I have raised the Soundness and Significance scores accordingly. I wish the authors the best of luck with their paper!

---

> > > ### Author Response · Authors · 2026-04-02
> > >
> > > Dear Reviewer MLWn,
> > >
> > > Thank you for your appreciation and encouragement! We are glad to have resolved your concerns and will certainly incorporate your constructive feedback into the final version of our paper.
> > >
> > > Best regards,
> > >
> > > Authors of FUSE

---

### Official Review · Reviewer_5SEN · 2026-03-13

**Soundness:** 2
**Presentation:** 3
**Significance:** 2
**Originality:** 2
**Overall Recommendation:** 4
**Confidence:** 3

**Summary:**

The paper introduces FUSE, a flow matching method for SBI that explicitly captures features of multi-modal inputs. Intermediate observation likelihoods are used to improve sample quality by guiding generative trajectories. This method achieves competitive performance compared to state-of-the-art baselines on common SBI benchmarks, and is shown to be effective on a real-world exoplanet orbital estimation task.

**Compliance With Llm Reviewing Policy:**

Affirmed.

**Final Justification:**

The authors addressed several of my initial concerns and provided additional empirical evaluations that further contextualize the proposed method among common baselines. While I believe a more comprehensive analysis of sample efficiency trade-offs is warranted in the revised manuscript, along with the full set of baseline methods on SBIBM, I'm encouraged by authors' engagement with feedback and now lean toward acceptance, improving my score from 3 to a 4.

**Key Questions For Authors:**

- Is there a viable extension of the proposed method to a sequential, non-amortized analog? Were any such formulations considered during method design or evaluation?
- In Figure 2, FUSE often has significantly worse C2ST scores compared to Simformer at the first reported simulation budget of 1,000 samples. Is there a particular reason why the proposed method may be expected to underperform on relatively small data, after which there is some crossover point where performance exceeds comparable methods?
- Was $\ell$-C2ST considered as an additional measure of posterior fidelity?
- Are the metric values reported in Table 1 averages taken at the $10^5$ simulation budget? I may have missed something in the appendix, but I couldn't specifics beyond those mentioned in Section 5.1, which states "all reported metrics are averaged." It was unclear to me whether this meant averages were taken at the largest simulation budget, or possibly across all simulation budgets, etc.

**Limitations:**

Yes

**Strengths And Weaknesses:**

**Strengths**
- The paper is well-organized, includes a comprehensive literature review, and provides high quality figures/diagrams.
- The paper's stated contributions are clear and attempt to address the difficult problem of efficiently modeling multimodal data in the SBI domain. Improving on the limitations of existing methods in this space presents an opportunity for high impact on tasks across many real-world domains, as many scientific inference problems involve observations from complex, multimodal structures.
- The proposed methodology is compelling and builds on recent advances in SBI, namely tokenizing inference data for use with transformer-based architectures (e.g., as with Simformer) and leveraging diffusion/flow-matching NDEs (e.g., as with FMPE) for improved accuracy and portability of learned models.
- Reported results appear competitive with relevant baselines on SBIBM tasks, and the real-world exoplanet case study provides a promising indicator of method stability on complex, higher-dimensional tasks.

**Weaknesses**
- The reported evaluations across SBIBM tasks would benefit from training methods over several trials and analyzing average-case performance (with error bars), as is common in SBI literature. Figure 2 and Table 1 appear to report individual training runs for each baseline, making it difficult to ascertain possible sensitivity to initialization, other factors during training, draws from the prior, etc. This would go some way toward verifying the stability of the method and the statistical significance of the reported metrics, where in some cases the proposed method offers a marginal performance advantage.
- In addition to the above remark, it would be insightful to include the FK-steering ablation, as reported for the orbital case study, alongside the primary SBIBM benchmark results. While the exoplanet task provides a promising indication that the FK guidance plays a meaningful role in posterior estimation, seeing results across the wide variety of dynamics captured by the SBIBM suite would provide further justification for the approach's design and/or highlight possible edge cases.
- The reported results on the core benchmark suite are missing key comparisons to relevant SBI methods, making the evaluation of the proposed method difficult to contextualize in the broader SBI/LFI landscape. In particular, methods like SMC-ABC often serve as a useful baseline reference, and score estimation methods like NPSE would complement the focus on flow-matching. Additionally, the positioning of the paper generally calls attention to sample efficiency, in which case sequential methods such as SNPE/SNLE/TSNPE/etc would likely provide meaningful reference points given their ubiquity in the SBI literature (even though the work focuses primarily on amortized methods).
- To the best of my knowledge, it is fairly uncommon to average results across all SBIBM tasks when focusing on method performance. The simulation environments across the SBIBM suite capture diverse dynamics, with some settings designed to be particularly difficult for posterior-direct methods (e.g., NPE), namely SLCP (i.e., "simple likelihood, complex posterior"). This makes fully averaged metrics somewhat difficult to interpret, at least across baselines like NPE, where environments like SLCP may have an outsized impact given their purposefully adversarial nature.
  It would additionally be insightful to see metric averages like those reported in Table 1 factored across each of the three simulation budgets. This would highlight possible tradeoffs across baselines between performance and simulation data dependence.
- Minor remarks:
  * On line 410 in the right-hand column, "minute" should be "minutes."
  * Punctuation is missing from the sentence ending in the right-hand column of line 65.

---

> ### Author Rebuttal · Authors · 2026-03-31
>
> We sincerely thank the reviewer for the thoughtful feedback, and recognition of our method's potential in tackling complex, multi-modal scientific inference problems. We have carefully addressed each of your concerns below, and we will incorporate these into our manuscript accordingly.
>
> **1. Clarification on Experimental Settings (W3 & Q3)**
>
> To clarify, Table 1 averages 10 SBIBM tasks at a $10^5$ budget. Both Table 1 and Figure 2 present the results of FUSE without FK-steering to ensure a fair, architecture-to-architecture comparison with the baselines. Additionally, our originally reported C2ST was computed per observation, so we have relabeled it to $\ell$-C2ST for clarity.
>
> **2. Evaluation of SBI benchmark (W1,3,4 & Q4)**
>
> We agree averaging hides task-specific nuances. Thus, we now report the results for each individual task separately, and will include standard deviation bars(across the observation sets) in Fig. 2. We also added SMC-ABC, NPSE, and sequential baselines; due to time limits, SNPE/SNLE ($10^5$) results use the original SBIBM benchmark (Lueckmann et al., 2021), with full evaluations deferred to the final manuscript. **Due to space limits, the table below shows an updated subset (SIR, SLCP, LV), we will report the full evaluation table including all ten tasks and all metrics with statstical analysis in the revised manuscript**
>
> |Method|SIR|SLCP|LV|
> |---|---|---|---|
> |FUSE w/o FK|0.55±0.01|0.63±0.05|0.85±0.09|
> |Simformer|0.69±0.05|0.72±0.05|0.90±0.05|
> |FMPE|0.60±0.04|0.88±0.04|0.92±0.05|
> |NPE|0.57±0.03|0.84±0.04|0.95±0.04|
> |SMC-ABC|0.62±0.03|0.96±0.02|0.995±0.003|
> |NPSE|0.57±0.02|0.69±0.10|0.87±0.07|
> |SNPE|0.55±0.02|0.67±0.06|0.93±0.08|
> |SNLE|0.58±0.03|0.58±0.03|0.70±0.10|
>
> Notably, while our amortized model is marginally behind sequential SNLE in C2ST, it crucially avoids SNLE's computationally prohibitive retraining per observation.
>
> **3. Performance at Low Simulation Budgets (Q2)**
>
> FUSE's underperformance at the $10^3$ budget (e.g., C2ST of 0.9997) inherently stems from its high-capacity Transformer (MMDiT) architecture, which requires more data to achieve optimal performance than light-weight models. However, as the budget scales to $10^5$, our base model still achieves a remarkable C2ST of 0.63, decisively surpassing all other amortized baselines.
>
> |Method \ Budget|10^3|10^4|10^5|
> |-|-|-|-|
> |FUSE w/o FK|0.9997 ± 0.0002|0.85±0.07|0.63±0.05|
> |Simformer|0.91±0.03|0.79±0.05|0.72±0.05|
> |FMPE|0.97±0.01|0.96±0.02|0.88±0.04|
> |NPE|0.95±0.02|0.89±0.04|0.84±0.04|
> |SMC-ABC|0.98±0.01|0.96±0.02|0.96±0.02|
> |NPSE|0.97±0.02|0.83±0.07|0.69±0.10|
> |SNPE|0.97±0.02|0.84±0.05|0.67±0.06|
> |SNLE|0.92±0.03|0.71±0.05|0.58±0.03|
>
> This table clearly demonstrates that while MMDiT is initially data-hungry, it possesses a significantly higher accuracy upper bound for large-scale, real-world scientific applications (e.g., the $10^7$ simulations in Figure 4). We will explicitly clarify this capacity-data tradeoff in the revised manuscript.
>
> **4. The Role and Ablation of FK-Steering on SBIBM Tasks (W2)**
>
> We introduced FK-steering primarily to tackle complex, non-Gaussian posteriors (e.g., SLCP, Exoplanet). For simpler SBIBM tasks, the base FUSE model already achieves near-optimal estimation, making steering gains marginal. To quantify the effect of FK-steering on complex environments like the SLCP task (Figure 3), we evaluate the 95% Credible Interval (C.I.) overlaps and the L2 distance between distribution peaks against gold-standard MCMC results.
>
> As shown in the table below, there is a fundamental trade-off between broad mass coverage and precise peak estimation. The unsteered model exhibits a broader posterior distribution, which achieves a higher interval overlap but suffers from a significantly worse L2 distance to the true peak. In contrast, FK-steering actively prunes physically implausible trajectories, steering the flow toward narrow, high-likelihood manifolds. This trades a slightly lower overlap rate for vastly superior peak accuracy.
>
> |Varitions|95% C.I. Overlap (Mean±Std) $\uparrow$|Peak L2 Dist. (Mean±Std) $\downarrow$|
> |:---|:---:|:---:|
> |FUSE|0.8390±0.1018|2.6410±1.6011|
> |FUSE w/o FK-Steering|0.8868±0.0368|3.7336±1.7577|
>
> We will include this full SBIBM ablation and trade-off discussion in the revised appendix.
>
> **5. Potential for Sequential / Non-Amortized Extensions (Q1)**
>
> We agree that extending FUSE to a sequential setting is entirely viable due to its modular design. First, the MM-DiT backbone is a versatile, general-purpose architecture for SBI that can be seamlessly integrated with sequential, non-amortized analog.
> Second, the FK-steering mechanism is particularly advantageous for sequential inference as it effectively prunes physically implausible regions. This significantly narrows the search space and accelerates the convergence of iterative posterior refinement. We consider this a highly promising future direction and will include a discussion of this potential in our conclusion.

---

> > ### Author Rebuttal · Reviewer_5SEN · 2026-04-04
> >
> > I appreciate the authors' detailed response and additional empirical evaluations. In particular, the factored task metrics across SBIBM tasks (in response to **Q-4**) are insightful, with comparisons to the competitive SNPE and SNLE baselines. Clarifications regarding reported simulation budgets and metrics (**W-3** and **Q-3**) are helpful for a more comprehensive interpretation of the results reported in the original manuscript.
> >
> > While the proposed method may attain competitive C2ST scores at larger simulation budgets, I believe the manuscript would benefit significantly from a more in-depth analysis of sample efficiency trade-offs. This is a key concern for many practitioners, and in many real-world settings, one may expect to have access to limited numbers of simulation samples (e.g., fewer than 10,000). Understanding how the proposed method fits into the larger SBI landscape at various budgets, e.g., as reported in response to **Q-2** for the SLCP task, would strengthen the positioning of the work, even if there are smaller budgets where the proposed method is outperformed.
> >
> > In total, I find the additional evaluations and engagement with reviewer feedback is encouraging, and am happy to improve my score. The full set of SBIBM tasks and performance (plus ablations, e.g., with and without steering) across different simulation budgets will improve the reader's understanding of the method's empirical impact, and I look forward to seeing these in the final manuscript.

---

> > > ### Author Response · Authors · 2026-04-07
> > >
> > > We sincerely thank you for your continued engagement, constructive feedback, and for improving your score. We completely agree that sample efficiency is a critical consideration for practitioners, and providing a transparent view of trade-offs at lower simulation budgets (e.g., < 10,000 samples) is essential for properly contextualizing our method. To fully address this, we will include a dedicated discussion in the "Limitations" section of the final manuscript, explicitly noting that while our high-capacity architecture achieves superior performance at larger budgets, it is more data-hungry and can be outperformed by lightweight or sequential baselines when budgets are strictly limited. Additionally, we will update the appendix to include the full suite of SBIBM tasks, reporting factored performance metrics and complete ablation studies (with and without FK-steering) across all evaluated simulation budgets. We deeply appreciate your time and effort, which has significantly strengthened the transparency and practical utility of our paper.

---

### Decision · Program_Chairs · 2026-04-30

**Decision:**

Accept (regular)

**Comment:**

This paper introduces FUSE, a method for simulation-based inference framework that combines a dual track diffusion transformer based on MM-DiT, with Feynman-Kac steered sampling. The method processes parameters and observations in distinct but interacting tracks. The approach is validated on the SBIBM benchmark and an application to orbital estimation.

The reviewers identified a number of strengths of this work. They found the architectural interpretation and framing compelling and well motivated, the results on the standard benchmark strong, and the paper well organized. They also raised a number of important concerns, which the authors addressed during the rebuttal phase:
* Initially there were missing common baselines as pointed out by reviewer 5sen. The authors added these in the rebuttal and the reviewer raised their score accordingly.
* Reviewer MLWn questioned the necessity of FK-steering. The authors added ablations and convinced the reviewer of its usefulness in this setting.
* Reviewers vqp1 and dyrw asked for additional discussion of the theoretical grounding of the FK-steering procedure. The authors clarified the theoretical usage. Reviewer dyrw did not acknowledge the discussion, and reviewer vqp1 was not entirely satisfied by the response finding issue with "what distribution the implemented discretized finite-particle sampler is actually approximating". I have looked carefully at the author's response and the reviewers' concerns. In my opinion I think this paper follows previous work in the sense that while the discretized finite particle sampler is difficult to characterize, at least the infinite particle continuous time limit is characterized.

Overall, I believe the authors have done a good job and offered a nice framing for a new work in the SBI literature. I recommend acceptance.